# StableRep: Synthetic Images from Text-to-Image Models Make Strong Visual Representation Learners

**Yonglong Tian**[1,*] **Lijie Fan**[1,2,*] **Phillip Isola**[2] **Huiwen Chang**[1] **Dilip Krishnan**[1]

[1]Google Research, [2]MIT CSAIL, *equal contribution

Code: https://github.com/google-research/syn-rep-learn

## Abstract

We investigate the potential of learning visual representations using synthetic images generated by text-to-image models. This is a natural question in the light of the excellent performance of such models in generating high-quality images. We consider specifically the *Stable Diffusion*, one of the leading open source text-to-image models. We show that (1) when the generative model is configured with proper classifier-free guidance scale, training self-supervised methods on synthetic images can match or beat the real image counterpart; (2) by treating the multiple images generated from the same text prompt as positives for each other, we develop a multi-positive contrastive learning method, which we call StableRep. With *solely synthetic* images, the representations learned by StableRep surpass the performance of representations learned by SimCLR and CLIP using the same set of text prompts and corresponding *real* images, on large scale datasets. When we further add language supervision, StableRep trained with 20M *synthetic* images achieves better accuracy than CLIP trained with 50M *real* images.

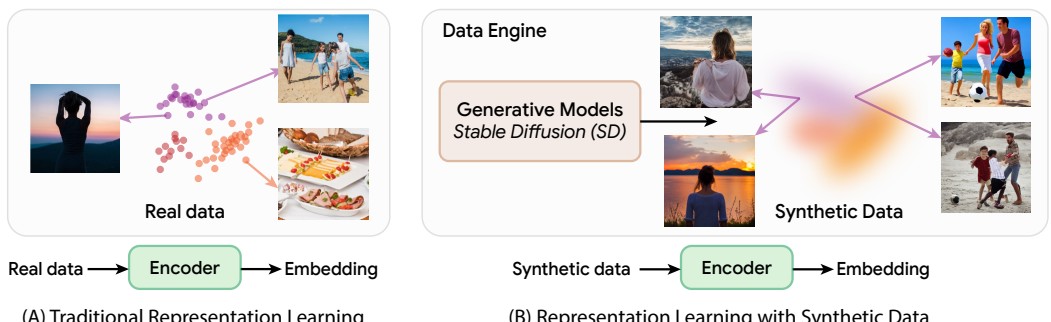

(A) Traditional Representation Learning  (B) Representation Learning with Synthetic Data

Figure 1: **Left:** traditional visual representation learning relies on a dataset of real images to train an image embedding function. **Right:** we view generative models as datasets that allow us to sample images from the data distribution. In our study, we leverage text-to-image models (Stable Diffusion [61]) and treat multiple images synthesized from the same prompt as positives for contrastive representation learning.

## 1   Introduction

*Data* has assumed a paramount role as the key component for the success of modern machine learning systems. Such systems, especially foundation models in various domains, heavily rely on vast and diverse datasets to acquire knowledge, make accurate predictions, and generate content. The quality, quantity, and diversity of the data significantly impacts the performance and effectiveness of these

37th Conference on Neural Information Processing Systems (NeurIPS 2023).

models, as they learn from the collective information encapsulated within the data. In this data-centric era, a central question is: how can we collect such large amounts of varied data to train AI models?

As an example, suppose we are trying to solve a new computer vision problem, and need to collect data (images) for it. An ideal situation is to place a camera anywhere in the wold and capture whatever we need. But in reality, collecting data is historically not easy. In the 1990s, researchers needed to take photos by themselves to create datasets for objects [52] and faces [68, 24]. To collect data in the 2000s, people crawled the Internet [15]. Noisy, uncurated data collected in such a manner can exhibit domain gaps with the real world problem and reflect imbalances due to societal bias. Removing or reducing such imperfection in data of high volume by human labeling is costly and can be prohibitive.

However, what if data collection could be simplified to the utterance of a natural language command, specifying what you want? What if, for hardly any cost, you could take a photo every few milliseconds? This sounds fanciful, but modern text-to-image generative models are approaching this vision. It has long been a dream that someday we could use these as our data sources, rather than taking photos [75, 30, 35]. In this paper, we study if this is now a practical option in the context of large scale visual representation learning.

To achieve this, we choose to work with Stable Diffusion [61], one of the leading open source text-to-image models. We synthesize images by prompting Stable Diffusion with text from large scale image-text datasets, such as CC12M [9] and RedCaps [16]. Surprisingly, our investigation reveals that when the classifier-free guidance scale is properly configured for Stable Diffusion, it is able to synthesize images on which training self-supervised methods can perform *at par with or better than* training on real images of the same sample size. Inspired by the idea of contrastive self-supervised learning, which promotes intra-image invariance, we develop a representation learning approach that promotes intra-caption invariance. We achieve this by treating the multiple images generated from the same text prompt as positives for each other and use them in a multi-positive contrastive loss (see Figure 1). Despite training with solely synthetic images, this approach, called StableRep, even outperforms state-of-the-art methods such as CLIP [58] using the same text set, but with corresponding real images, on various representation evaluation benchmarks.

Intuitively, one reason that synthetic data can be better than real data is because we are able to achieve a greater degree of control in the sampling, such as via the guidance scale in Stable Diffusion, or via text prompts and latent noise variables. Furthermore, generative models have the potential to generalize beyond their training data and therefore provide a richer (synthetic) training set than the corresponding real data alone. Our key contributions are:

1. We discover that training modern self-supervised methods on synthetic images from Stable Diffusion can be surprisingly effective. The learned representations are often better than representations learned from real images of the same sample size.
2. We develop StableRep, a novel representation learning approach by capturing invariance between images generated from the same text prompt, and propose a multi-positive contrastive loss.
3. With StableRep, we are able to achieve 76.7% linear accuracy on ImageNet with ViT-B/16, using solely synthetic images.
4. When coupled with language supervision, our StableRep trained with 20M *synthetic* images (10M captions) achieves better accuracy than CLIP trained with 50M *real* images (50M captions).

## 2   Standard Self-supervised Learning on Synthetic Images

A typical visual representation learning algorithm takes an image *dataset* $\{\mathbf{x}_i\}_{i=1}^N$ as input, and yields an image encoder $F : \mathbf{x} \to \mathbf{e}$, which embeds an image $\mathbf{x}$ into a vector $\mathbf{e}$. In this paper, we instead try to produce a good $F$ by using a *generative model $G$* rather than a real image *dataset*. Specifically, we focus on text-to-image generative models $G : (\mathbf{t}, \mathbf{z}) \to \mathbf{x}$, which maps a pair of text $\mathbf{t}$ and latent noise $\mathbf{z}$ to an image $\mathbf{x}$. While there are several top performing text-to-image models [59, 67, 88, 7, 36, 3], we conduct our exploration with the Stable Diffusion [61] since it is publicly available and widely used. The version we used is v1-5.

### 2.1   Synthetic images from Stable diffusion

Stable diffusion [61] is a denoising diffusion probabilistic model [73, 31] that runs the diffusion process in the latent space of an autoencoder. It improves the sample quality and text-image alignment

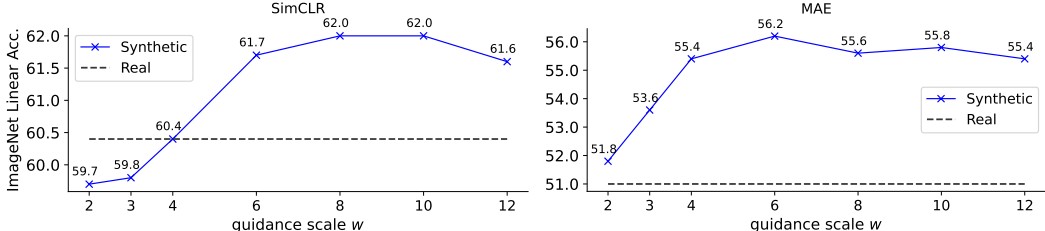

Figure 2: Performance of linear probes on ImageNet as a function of the guidance scale of Stable Diffusion generation. **Left**: using SimCLR as pre-training; **Right**: using MAE as pre-training. In both cases, we see pre-training on synthetic images that are generated by Stable Diffusion with a guidance scale between 6 and 8, gives a significant boost over training only on real images. We used the CC3M dataset for these experiments.

via classifier-free guidance [32], which linearly combines conditional score estimate $\epsilon(\mathbf{t}, \mathbf{z}_\lambda)$ and unconditional estimate $\epsilon(\mathbf{z}_\lambda)$ with the guidance scale $w$ at each step $\lambda$:

$$\tilde{\epsilon}(\mathbf{t}, \mathbf{z}_\lambda) = w\epsilon(\mathbf{t}, \mathbf{z}_\lambda) + (1 - w)\epsilon(\mathbf{z}_\lambda) \tag{1}$$

The Stable Diffusion model $G_{sd}$ relies on text sources to generate images. Instead of collecting a corpus of captions from scratch, we use the text part of existing *uncurated* image-text pair datasets, such as CC3M [71] and CC12M [9]. Formally, given an image caption dataset $\{\mathbf{t}_i\}_{i=1}^N$, we generate one image per caption, forming a synthetic image dataset of the same size.

## 2.2 Self-supervised learning on synthetic images

Recent representative self-supervised learning algorithms are mostly from two families: (1) contrastive learning which encourages invariance between embeddings of different augmentations of the same image ; (2) masked image modeling where model uses unmasked patches to predict masked patches (although there are other methods that fall into neither category, such as BYOL [25] and DINO [6]). For our study, we choose SimCLR [10] from the former family and MAE [26] from the latter due to their simplicity and strong performance. We focus on the Vision Transformer architecture [18], and use captions from CC3M [71] except when noted.

**SimCLR [10].** We directly train SimCLR with ViT-B/16 on the synthetic image dataset, and measure the representation quality by linear probing evaluation on ImageNet [15] [1]. One factor to consider is the classifier-free guidance scale $w$, as it trades off between diversity and quality of the synthesized images and thus can affect the learned representations. To study this, for each $w$ in the set $\{2, 3, 4, 6, 8, 10, 12\}$, we generate a copy of size $N$ (one image per caption) to train SimCLR. Figure 2(left) visualizes the influence of $w$. The optimal $w$ is around 8 (both 8 and 10 give an accuracy of 62.0%). This is different from the FID metric where $w = 2$ is the optimal.

The captions $\{\mathbf{t}_i\}_{i=1}^N$ used to generate synthetic images are also paired with $N$ real images. We train a SimCLR model with these real images. This model achieves 60.4% accuracy, experiencing a 13% drop in linear accuracy compared to pre-training on ImageNet. Such gap has been generally observed for uncurated pre-training data [77]. However, both interestingly and surprisingly, synthetic images with $w = 8$ have 1.6% higher accuracy than real images (62.0% v.s. 60.4%).

**MAE [26].** Following the default hyperparameters in MAE [26], we train a ViT-B/16 model for each guidance scale $w$. Figure 2(right) reports the linear probing results. The accuracy of synthetic images increases quickly with $w$ after 2, and gradually drops when $w$ is large, e.g., $w \geq 10$. The optimal guidance scale for MAE is 6, and this is different from SimCLR where the accuracy peaks at 8 or 10. This suggests that different methods may require different $w$. With $w = 6$, synthetic images have a 4.2% better accuracy than real images.

While the linear probing accuracy of MAE is lower than that of contrastive methods, its effectiveness often comes with fine-tuning. When fine-tuning pre-trained MAE models on ImageNet, we found synthetic images are still able to outperform real images. For instance, synthetic images with $w = 6$ is 0.3% higher than real images (82.9% v.s. 82.6%).

---

[1]We verify our SimCLR implementation by pre-training on ImageNet. It achieves 74.3% linear probing accuracy. As a comparison, SimCLR in [11] with the same architecture and epochs achieved 73.9%.

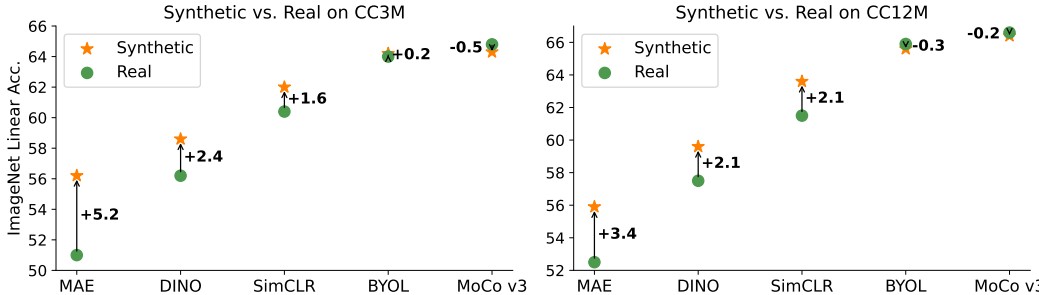

Figure 3: Training self-supervised methods on synthetic images can be better than, or on par with, real images of the same sample size. **Left**: CC3M dataset; **Right**: CC12M dataset
.

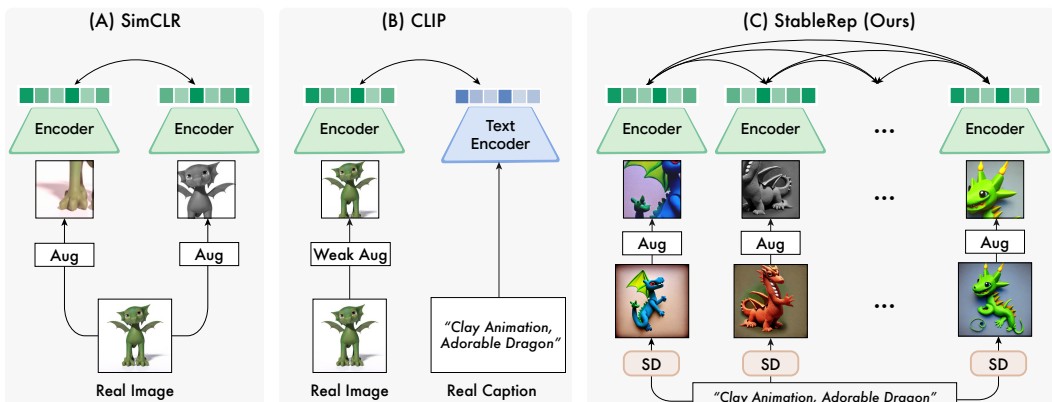

Figure 4: We compare our pipeline (C) to that of (A) SimCLR; (B) CLIP. In SimCLR, the real image is augmented to give two views which are contrasted against each other through the same encoder. For CLIP, a real image and corresponding real caption are passed into image and text encoder, the image is augmented (usually more weakly than for SimCLR) followed by a contrastive loss. In our pipeline, each real caption is passed into Stable Diffusion (SD) to generate a number of synthetic images. These synthetic images are then augmented as in SimCLR, and treated as positives for each other in a multi-positive contrastive loss.

**Other SSL methods.** To test if synthetic images can be generically applied to different self-supervised learning methods, we try three more representative approaches: BYOL [25], MoCo-v3 [11], and DINO [6]. We do not tune $w$ for each method, and instead apply the optimal $w$ (= 8) discovered for SimCLR. The results on CC3M and CC12M are visualized in Figure 3. Synthetic images significantly improve over real for MAE, DINO, and SimCLR, and performs on par with real for BYOL, and slightly worse for MoCo-v3 (which could be attributed to not tuning the guidance scale $w$).

## 3   Multi-Positive Contrastive Learning with Synthetic Images

Text-to-image generative models offer a new way to compose positive samples for contrastive learning. Given an image caption, we can create multiple diverse samples by starting the reverse diffusion process with different latent noise $\mathbf{z}$. Since these images are produced using the same prompt, they possess similar visual semantics, making them suitable for use as multiple positive samples for each other in contrastive learning. This property is unique to generative models, since collecting multiple images for each caption in large scale is infeasible. Figure 4 compares our StableRep pipeline with that of SimCLR and CLIP.

**Multi-positive contrastive loss.** We describe multi-positive contrastive learning as a matching problem. Consider an encoded anchor sample $\boldsymbol{a}$, and a set of encoded candidates $\{\boldsymbol{b}_1, \boldsymbol{b}_2, ..., \boldsymbol{b}_K\}$.

We compute a contrastive categorical distribution $\mathbf{q}$ that describes how likely $\boldsymbol{a}$ is to match each $\boldsymbol{b}$:

$$\mathbf{q}_i = \frac{\exp(\boldsymbol{a} \cdot \boldsymbol{b}_i/\tau)}{\sum_{j=1}^{K} \exp(\boldsymbol{a} \cdot \boldsymbol{b}_j/\tau)} \tag{2}$$

where $\tau \in \mathcal{R}_+$ is the scalar temperature hyper-parameter, and $\boldsymbol{a}$ and all $\boldsymbol{b}$ have been $\ell_2$ normalized. Intuitively, this is a $K$-way softmax classification distribution over all encoded candidates. Assume there is at least one candidate that the anchor $\boldsymbol{a}$ matches. Then we know the ground-truth categorical distribution $\mathbf{p}$ is:

$$\mathbf{p}_i = \frac{\mathbb{1}_{\text{match}(\boldsymbol{a},\boldsymbol{b}_i)}}{\sum_{j=1}^{K} \mathbb{1}_{\text{match}(\boldsymbol{a},\boldsymbol{b}_j)}} \tag{3}$$

where the indicator function $\mathbb{1}_{\text{match}(\cdot,\cdot)}$ indicates whether the anchor and candiate match. Then the multi-positive contrastive loss is the cross-entropy between the ground-truth distribution $\mathbf{p}$ and the contrastive distribution $\mathbf{q}$:

$$\mathcal{L} = H(\mathbf{p}, \mathbf{q}) = -\sum_{i=1}^{K} \mathbf{p}_i \log \mathbf{q}_i \tag{4}$$

This is a generalized form of the widely-used single-positive contrastive loss [54], where $\mathbf{p}$ reduces to a one-hot vector. This loss is closely related to that in [39], but a key distinction here is that we have no image class labels, and only assume images generated from the same caption are matched.

The PyTorch-like pseudocode of the batched multi-positive contrastive learning algorithm is described in Algo. 1. Each batch consists of $n * m$ images, meaning that we sample $m$ images for each of the $n$ captions. Here we still apply data augmentation, even though images from the same caption are different. This is to reduce overfitting since we perform many epochs of training over pre-generated synthetic images. However, if in the future the image generator is capable of producing images fast enough, then we can draw batches online and data augmentation may not be necessary. The

---

**Algorithm 1** Multi-Pos CL: PyTorch-like Pseudocode

```
# f: encoder: backbone + proj mlp
# tau: temperature

# minibatch x: [n, m, 3, h, w]
# n captions, m images per caption
for x in loader:
    x = augment(x)
    x = cat(unbind(x, dim=1)) # [n*m, 3, h, w]
    h = f(x)

    # compute ground-truth distribution p
    label = range(n * m) % n
    p = (label.view(-1, 1) == label.view(1, -1))
    p.fill_diagonal(0) # self masking
    p /= p.sum(1)

    # compute contrastive distribution q
    logits = h @ h.T / tau
    logits.fill_diagonal(-1e9) # self masking
    q = softmax(logits, dim=1)

    H(p, q).backward()

def H(p, q): # cross-entropy
    return - (p * log(q)).sum(1).mean()
```

**Notes**: `h.T` is h's transpose. The $\ell_2$ normalization operator is included in the encoder `f`.

---

multi-positive contrastive learning algorithm is also generic such that SimCLR can also be described by it – we begin by randomly selecting a set of $n$ images and subsequently apply $m$ (set as 2) crops to each of the chosen images. However, in our StableRep we only utilize a single crop from each image.

# 4 Experiments

We perform StableRep pre-training on synthetic images synthesized from texts in the CC3M (2.7 million samples) [71], CC12M (10 million) [9], or RedCaps datasets (11.6 million) [16]. We then evaluate the frozen representations by (1) linear probing on ImageNet-1k and other smaller scale image classification benchmark, and (2) few-shot image recognition that measures the generalization ability of the representations.

**Backbone.** We use ViT models [18] as the backbone for our approach StableRep. On top of the `CLS` token, we apply a 3-layer MLP projection head with hidden layers of 4096 dimensions and an output of 256 dimensions. Batch Normalization [33] is used in this projection head.

**Training.** In most of our experiments, we adopt a batch size of 8192 images (i.e. $m * n = 8192$). This way the computation of each batch is equivalent to SimCLR with a batch size of 4096, because each image in SimCLR has two crops. We use AdamW optimizer [46] with a learning rate of 0.0032 and weight decay of 0.1, and set $\beta_1, \beta_2$ as $0.9, 0.98$ respectively. We pre-generate 10 images for each text prompt. In each iteration, we randomly sample 6 out of the 10 for each sampled caption to form the training batch, i.e., $m = 6$ in Algo. 1. Recall that for SimCLR $m = 2$. As a result, one epoch training of StableRep is computationally equivalent to 3 epochs of SimCLR. To provide easy comparison, we report SimCLR-equivalent epochs for StableRep in all of our analysis.

## 4.1 Main results on CC12M and RedCaps

In this section, we perform StableRep on images synthesized by either CC12M or RedCaps. For StableRep, we first removed duplicate captions from each dataset, resulting in a reduced number of captions: from 10.0M to 8.3M for CC12M and from 11.7M to 10.5M for RedCaps. We compared StableRep to SimCLR, which was trained on either synthetic or original real images. We also included CLIP with a synthetic and a real version [2]. For SimCLR and CLIP, we did not perform de-duplication for either real or synthetic setting. We train for 35 epochs for all methods using ViT-B/16 (for StableRep, this refers to 35 SimCLR-equivalent epochs). We observed that CLIP started to overfit around 30 epochs. But StableRep did not overfit with this schedule (see Table 6c for results with longer training). For StableRep, we additionally apply random downsample augmentation (see Appendix A.1 for details and how such downsample affects different methods).

**ImageNet.** Table 1 presents the results of linear probing on ImageNet. For StableRep, we prepend a BatchNorm layer without affine transformation to the linear classifier (see Appendix A.5 for more details). We observed that training SimCLR on synthetic images yields an improvement of 2.2% top-1 accuracy on CC12M and 1.0% on RedCaps when compared to real images. However, the accuracy of CLIP drops by 2.6% on CC12M and 2.7% on RedCaps when trained on synthetic images (see Section 5 for more discussion). On the other hand, our method StableRep outperforms CLIP trained on real images, with improvements of 3.2% and 2.6% for CC12M and RedCaps, respectively.

| | Real | | Syn | | | | Real | | Syn | | |
| | SimCLR | CLIP | SimCLR | CLIP | StableRep | | SimCLR | CLIP | SimCLR | CLIP | StableRep |
|---|---|---|---|---|---|---|---|---|---|---|---|
| acc. | 61.5 | 70.3 | 63.7 | 67.8 | 73.5 | acc. | 61.8 | 71.9 | 62.8 | 69.2 | 74.5 |

(a) CC12M          (b) RedCaps

Table 1: Comparison under the linear probing protocol on ImageNet [15]; measuring top-1 accuracy on a frozen pre-trained backbone. We compare our StableRep with SimCLR [10] and CLIP [58] with either synthetic or real images, on CC12M [9] and RedCaps [16]. All models are pre-trained with 35 epochs using ViT-B/16 [18].

**Linear classification on more datasets.** We followed the approach of SimCLR [10] and BYOL [25] to assess the generality of our learned representations across different image domains. Specifically, we performed linear classification on 11 image classification datasets introduced by [40]. The results are reported in Table 2, and the relative performance is consistent with that on ImageNet. Notably, our proposed method, StableRep, achieves the highest accuracy on all of the 11 datasets.

| | | CIFAR-10 | CIFAR-100 | Aircraft | Cars | DTD | Flowers | Pets | SUN397 | Caltech-101 | Food-101 | VOC2007 | Average |
|---|---|---|---|---|---|---|---|---|---|---|---|---|---|
| Real | SimCLR | 88.3 | 70.3 | 47.1 | 45.5 | 76.2 | 92.5 | 70.1 | 65.4 | 83.8 | 75.0 | 81.2 | 72.3 |
| | CLIP | 94.0 | 79.0 | 53.2 | 75.8 | 75.7 | 96.0 | 86.7 | 72.5 | 92.7 | 81.6 | 86.1 | 81.2 |
| Syn | SimCLR | 84.8 | 65.2 | 51.0 | 53.2 | 74.5 | 93.3 | 74.2 | 65.0 | 81.7 | 74.8 | 81.8 | 72.7 |
| | CLIP | 87.3 | 69.5 | 53.5 | 79.5 | 75.8 | 95.4 | 85.8 | 69.2 | 90.9 | 78.3 | 84.5 | 79.1 |
| | StableRep | **96.2** | **84.1** | **58.3** | **80.9** | **78.1** | **97.2** | **87.5** | **73.0** | **94.6** | **83.6** | **87.2** | **83.7** |

Table 2: Linear probing experiments on image datasets from various domains. Pre-training is conduceted on CC12M, with either synthetic or real images. Best results for each dataset are highlighted with **bold**.

---

[2] We verified our CLIP implementation by comparing to prior work [51] on CC12M. With ViT-B/16, our CLIP achieved 40.2% zero-shot and 70.3% linear accuracy on ImageNet (*v.s.* 36.0% and 69.0% in [51]).

| | | CIFAR-10 | CIFAR-100 | Aircraft | Cars | DTD | Flowers | Pets | SUN397 | Caltech-101 | Food-101 | Average |
|---|---|---|---|---|---|---|---|---|---|---|---|---|
| Real | SimCLR | 64.0 | 70.4 | 40.7 | 50.9 | 82.2 | 92.1 | 74.4 | 94.0 | 90.4 | 70.4 | 73.0 |
| | CLIP | 77.5 | 82.1 | 62.0 | 90.9 | 83.3 | 97.6 | 91.1 | **97.2** | 98.2 | 87.0 | 86.7 |
| Syn | SimCLR | 50.0 | 58.9 | 45.2 | 54.2 | 79.8 | 92.0 | 74.6 | 92.9 | 89.1 | 71.0 | 70.8 |
| | CLIP | 63.1 | 73.5 | 61.3 | **92.5** | 81.7 | 96.9 | 91.5 | 96.7 | 96.8 | 82.5 | 83.7 |
| | StableRep | **92.3** | **91.8** | **62.6** | 91.8 | **86.4** | **98.2** | **91.7** | **97.3** | **98.8** | **87.3** | **89.8** |

Table 3: Few-shot experiments. We report 5-way, 5-shot classification performance. Best results for each dataset are highlighted with **bold**.

**Few-shot image classification.** Prior work [80, 83, 17] has shown that representation learning is the key for few-shot image classification. A simple classifier on top of frozen representation is sufficient to achieve strong results. We perform 5-way, 5-shot classification following the setup in [83, 19]. As shown in Table 3, StableRep stands out on 9 out of the 10 datasets.

| | MAE | StableRep | | | |
|---|---|---|---|---|---|
| | IN1k, Real | cc12m, 35ep | cc12m, 105ep | redcaps, 35ep | redcaps, 105ep |
| mIoU | 48.1 | 48.8 | **49.4** | 47.3 | 48.4 |

Table 4: ADE20k semantic segmentation (mIoU) using UperNet. StableRep models are trained by 35 or 105 SimCLR-equivalent epochs.

**Semantic segmentation.** We fine-tune pre-trained StableRep models on ADE20k [91] using Uper-Net [86]. For this evaluation, StableRep is pre-trained for 35 or 105 epochs. Table 4 shows that StableRep trained on synthetic data is able to outperform MAE trained on the real ImageNet images, despite StableRep has no masked image modeling which benefits dense prediction tasks.

## 4.2 Ablation analysis

For simplicity, ablation studies in this section do not use the random downsample augmentation in pre-training or prepend an extra BatchNorm layer to the linear classifier.

The design choice of $m$ (number of synthetic images per caption) is one of the key design choices for our approach. Therefore we study the following two factors relevant to $m$ on CC3M captions (2.7 million after de-duplication).

| $l$ | 1 | 2 | 4 | 6 | 8 | 10 |
|---|---|---|---|---|---|---|
| acc. | 61.2 | 64.2 | 65.6 | 66.0 | 66.2 | 66.2 |

| $m$ | 1 | 2 | 4 | 6 | 8 | 10 |
|---|---|---|---|---|---|---|
| acc. | 60.5 | 68.7 | 69.6 | 69.6 | 69.8 | 69.5 |

(a) Given a generation budget $T$, we use $T/l$ captions and generate $l$ images per caption. When $l = 1$, we train a SimCLR model.

(b) Given a batch size of $C$, we form each batch by sampling $C/m$ captions and $m$ images per caption. We "abuse" $m = 1$ to represent SimCLR.

Table 5: **Ablation experiments on CC3M.** ImageNet linear probing results with design choices relevant to data generation parameter $l$, and batch sampling parameter $m$.

**Fixed generation budget.** We first study the question: given a fixed value for the number of total synthetic images generated ($T$), should we generate more images per caption ($l$), and therefore use fewer captions ($T/l$) or the reverse. We assume an image budget of $T = 2.7$ million. During training, we use the same total batch size (8192) for all $l$, and set the sampling parameter $m$ as $l$. Table 5a presents the results. There is a clear benefit of generating more than 1 image per caption, e.g., $l = 8$ improves over $l = 1$ by 4.8%. But this benefit saturates around $l = 10$. We thus generate 10 images per caption for our final experiments.

**How to form the batch.** Suppose we have generated 10 images for each of the 2.7 million captions. Now given a fixed batch size, i.e., $n * m = C$ (recall that $n$ is the number of captions, $m$ is the number of images per caption, inside each batch), a larger $m$ encourages stronger invariance of images from the same caption, while larger $n$ incorporates more negatives and thus encourages better separability

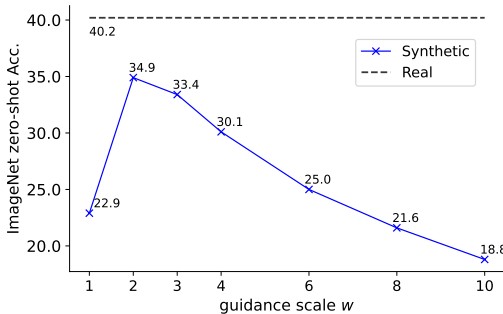
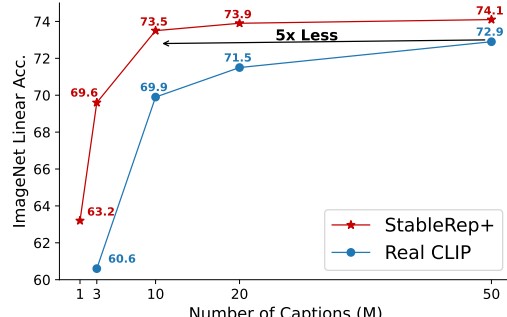

Figure 5: ImageNet zero-shot accuracy with different Stable Diffusion generation guidance scale $w$, using CLIP as pre-training.

Figure 6: ImageNet linear probing accuracy comparison between StableRep+ on synthetic images and CLIP on real images on LAION subsets. For this experiment, only 2 images are generated for each caption.

of representations. To study this trade-off, we vary the sampling parameter $m$ from 2 to 10 while keeping $n = C/m$. As shown in Table 5b, The linear probing accuracy are similar between $m = 4$ and $m = 10$ (peak at $m = 8$ with 69.8% accuracy), showing the robustness of StableRep w.r.t. $m$. We choose $m = 6$ as our default setup. We "abuse" $m = 1$ to represent SimCLR.

After the above study, we continue to ablate the following factors on CC12M and RedCaps.

| Case | IN | avg. |
|---|---|---|
| small | 72.8 | 82.2 |
| large | 70.8 | 80.4 |
| mixed | 71.9 | 81.5 |

| Size | IN | avg. |
|---|---|---|
| ViT-B/16 | 72.8 | 82.2 |
| ViT-L/16 | 74.7 | 82.9 |

| Epochs | CC12M | RedCaps |
|---|---|---|
| 35 | 72.8 | 73.7 |
| 70 | 75.0 | 76.3 |
| 105 | 75.7 | 76.7 |

(a) **Guidance scale** $w$. Smaller $w$ yields better linear accuracy.

(b) **Model size.** Our approach scales up with model size.

(c) **Training epochs.** Longer training further improves accuracy.

Table 6: **Ablation experiments** by pre-training on CC12M or RedCaps. We report linear probing accuracy on ImageNet (IN) and/or average accuracy over the 11 fine-grained classification datasets (avg.). The colored cell indicates the default setup on each dataset: ViT-B/16 trained for 35 epochs with small guidance scale $w$.

**Guidance score for training.** We consider three configurations for the classifier free guidance scale $w$: (1) *large scale* $- w \in \{8, 10\}$; (2) *small scale* $- w \in \{2, 3\}$; (3) *mixed scale* $- w \in \{2, 3, 4, 5, 6, 8, 10, 12\}$. As shown in Table 6a, small scale gives the best linear transfer accuracy on ImageNet and fine-grained classification datasets. This is possibly because smaller $w$ leads to larger intra-caption variation between generated images, which enforces StableRep to learn stronger invariance. This is different from SimCLR which requires larger $w$ (recall Section 2.1), as SimCLR only models intra-image invariance and thus higher image quality (larger $w$) helps more.

**Model scale.** We switch the backbone architecture to ViT-L/16. Table 6b presents the results. The accuracy improves by 1.9% on ImageNet linear probing and 0.7% on the average over fine-grained classification datasets. We found that pre-training with ViT-L was unstable. The loss kept exploding to `NaN`, and we resumed from the checkpoint before `NaN`. But this led to a higher convergent loss than ViT-B (ViT-L loss is lower before exploding). This may partly be due to the usage of BatchNorm.

**Longer training.** To investigate the scaling behavior of StableRep w.r.t. training compute, we further increase the pre-training computation budget to 2x and 3x epochs, and report the linear probing accuracy on ImageNet in Table 6c. The results indicate that StableRep scales well with longer training, e.g., improving by 2.2 for 2x and 2.9 for 3x on CC12M pre-training, and by 2.6 for 2x and 3.0 for 3x on RedCaps pre-training.

## 5 Adding Language Supervision

How would training CLIP using synthetic images work? We study this question by generating a copy (one image per caption) for each guidance scale $w$ in $\{1, 2, 3, 4, 6, 8, 10\}$ and training CLIP using

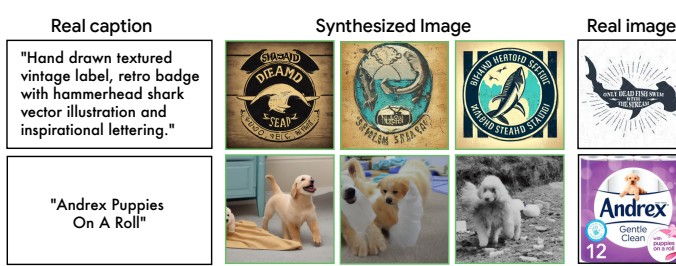

Figure 7: Examples of misalignment between input text and synthesized image, which can lead to suboptimal performance for CLIP trained on synthetic images. **Upper:** require headhammer shark but Stable Diffusion often generates sharks without headhammer; **Lower:** "Andrex Puppies" is a brand of toilet rolls.

each copy. Figure 5 plots the zero-shot ImageNet accuracy. Contrary to SSL methods, CLIP favors lower $w$. With the optimal $w = 2$, CLIP achieves 34.9% zero-shot accuracy. This is 5.4% lower than training on real images (40.2%). Such gap may be explained by misalignment between the generated images and the input text, shown in Figure 7. This is especially true for fine-grained classes.

We can add language supervision to StableRep by adding $0.5 * (\mathcal{L}_{i2t} + \mathcal{L}_{t2i})$ to StableRep loss, where $\mathcal{L}_{i2t}$, $\mathcal{L}_{t2i}$ are image-to-text and text-to-image contrastive losses described by Eq. 4. Adding supervision improves StableRep from 72.8% to 74.4% on CC12M and from 73.7% to 75.4% on RedCaps for ImageNet linear probing. We term it as StableRep+. We then further scale StableRep+ to a randomly selected 50M subset of LAION-400M [70]. For this experiment, we only generate 2 images per caption with $w = 2$, and train CLIP with *real* images and StableRep+ with *synthetic* images using different scales of random subsets of the 50M data. We plot the results in Figure 6. StableRep+ consistently achieves better accuracy than CLIP. Notably, StableRep+ with 10M captions outperforms CLIP with 50M captions, yielding a 5x time caption efficiency (2.5x image efficiency).

## 5.1 Fairness and compositionality

We further study the fairness and compositional understanding of the learned models on FairFace [37] and ARO [89] benchmarks, respectively. The results are presented in Table 7.

| | | | FairFace | | | ARO |
|---|---|---|---|---|---|---|
| | | pre-train data | mean acc. | best-class acc. | worst-class acc. | relation acc. |
| cc12m | CLIP | Real | 28.2 | 60.2 | 0.3 | 46.4 |
| | | Syn | 30.4 | 64.0 | 3.1 | **50.0** |
| | StableRep+ | Syn | **37.2** | **74.9** | **10.0** | 47.3 |
| redcaps | CLIP | Real | 9.3 | 31.1 | 0.4 | **59.0** |
| | | Syn | 22.3 | 52.4 | 1.0 | 56.0 |
| | StableRep+ | Syn | **27.3** | **64.4** | **2.1** | 52.3 |

Table 7: Results of fairness and compositionality evaluation.

**Fairness.** We perform zero-shot classificaton on FairFace. We jointly classify both races and genders, e.g., treating *Black male*, *Black female*, *Indian female*, and so on as different classes at the same time. For cc12m models, CLIP with real data only achieved 0.3% accuracy with *Southeast Asian male* class, and CLIP wth synthetic data improves this class to 3.1%, while our StableRep+ furthers it to 27.2%. For redcaps models, real CLIP only has 0.4% accuracy for *East Asian Male*, while StableRep+ improves this class to 22.8%. In summary, training with synthetic data is able to improve the worst class accuracy. However, a obvious geographic bias still exists in all models.

**Compositionality.** The results of compositionality evaluation are less clear. While training with synthetic data on cc12m slightly improves the relational understanding, an accuracy drop is observed in models trained with synthetic data on redcaps. An in-depth investigation may be further needed.

## 6 Related Work

**Text-to-Image generative models.** Text-to-image models trained on large image and text pairs have recently enabled the creation of rich and diverse images encompassing many genres and themes [7, 61, 67, 88]. The resulting creations have become a sensation, with Stable Diffusion having

millions of downloads and many tools for image manipulation built on top [66, 38, 90]. Most of these models are built on denoising diffusion models [31, 73] with some notable exceptions [8, 7]. In this paper, we leverage this latest generation of diffusion-based pre-trained generative models for the task of representation learning.

**Visual representation learning.** Early approaches for visual representation learning often relied on pretext tasks such as inpainting [56] to train image encoders. More recent advancements have shown that mask image modeling, a form of self-supervised training, can be highly effective. In particular, Masked Autoencoder (MAE) [26] has demonstrated significant improvements in downstream fine-tuning performance. Another line of research focuses on contrastive learning, which aims to learn visual representations by maximizing agreement between two augmented views of the same image while distinguishing it from negative examples [10, 78, 54, 84, 27, 79]. Meanwhile CLIP [58] and its subsequent works [51] leverage contrastive learning to train image representations using language supervision, leading to impressive transferability across various tasks.

**Learning from synthetic data.** It has been common to train machine learning models with synthetic data in different domains [72, 81, 14, 63, 44, 64, 50, 43, 76, 87, 29, 49]. In computer vision, synthetic images have been used as a source for training models, such as optical flow [48, 23], autonomous driving [1], semantic segmentation [12, 62], object detection [65, 57], human pose estimation [82, 34] or classification [2, 69, 28]. The closest set of work are the ones that conduct representation learning on synthetic images [60, 45, 4, 35]. In [60], a model is trained to perform multi-task learning on synthetic images. The main method in [45, 4, 35] is to manipulate the latent variable of deep generative models [45, 4, 35] or image generation procedures [4], to form meaningful synthetic images for their representation learning methods. Our method falls into this category, but we use text-to-image diffusion models, which have also been explored by [2, 28, 69]. The key difference is that they conducted supervised learning while we use synthetic data for pre-training representations.

# 7 Conclusion, Limitations and Broader Impact

We have shown that solely synthetic data generated from state of the art text-to-image models can be used to train powerful visual representations. By harnessing the stochastic nature of Stable Diffusion in combination with a multi-positive contrastive loss, our approach yields a representation that surpasses the performance achieved through training on real data alone. Through a series of experiments, we establish that pre-training with synthetic datasets of varying scales yields impressive results across different downstream tasks, including linear probing and few-shot classification. Interestingly, we discover that even vanilla self-supervised methods trained on synthetic data can either outperform or achieve comparable results to those trained on real data.

Despite demonstrating the potential of training with synthetic data, this paper acknowledges its limitations. Firstly, we have yet to comprehend the reasons behind the effectiveness of training self-supervised methods on synthetic images compared to an equal amount of real images. It is possible that this observation is confined to our particular evaluation methodology. Furthermore, the current image generation process remains slow, with approximately 0.8s per image on a A100 GPU or 2.2s per image on a V100 GPU while xFormers is enabled. Consequently, we are not able to train StableRep models with non-repetitive images synthesized online. Additionally, we have not addressed the issue of semantic mismatch between the input prompts and the generated images, which may impact the quality and usefulness of the synthetic data. Moreover, synthetic data has the potential to exacerbate biases due to mode collapse and a predisposition to output "prototypical" images. Lastly, image attribution becomes a challenge when working with synthetic data.

**Broader impacts.** This paper focuses on the fundamentals of visual representation learning, and we believe it will be beneficial to the practice of this field. Our method presents an immediate application by reducing the reliance on collecting a vast amount of real images for learning representations. This approach brings potential benefits in terms of cost-effectiveness and minimizing biases introduced through human collection and curation processes. However, it is important to acknowledge that our method relies on text-to-image generative models trained on large-scale, uncurated web data. Such data may conceal social biases and errors that would have been exposed through human curation. Additionally, we must recognize that the text prompts we employed are not completely bias-free; the selection of prompts influences the synthesized images. Thus, the choice of prompts assumes a role similar to the selection of real images for self-supervised visual representation learning.

## Acknowledgements

We would like to thank anonymous reviewers and Shumeet Baluja for reviewing our manuscript and providing many helpful comments and suggestions. We also appreciate the helpful discussions and the general supports from the VisCam teammates in Google Research.

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

# A  Implementation Details

## A.1  Downsample augmentation

Synthetic images have constant high resolutions (e.g., 512×512 for Stable Diffusion). We find this leads to a domain gap when transferring to situations involving low resolution images, such as CIFAR-10 or CIFAR-100. To address this issue, we introduce `Random Downsample` augmentation, which randomly resizes images to a resolution of 64 or 128 (equally probable) and then resizes them back to 224. During pre-training, we apply this augmentation with a probability of 0.05 and prepend it to other augmentations.

In Table 8, we ablate the effects of applying this random downsample augmentation to different pre-training methods. This augmentation brings significant improvements on CIFAR-10 and CIFAR-100 datasets, while maintaining the performance on other datasets. On average this augmentation is more beneficial for pre-training with synthetic images than real ones.

| | | Downsample | CIFAR-10 | CIFAR-100 | Aircraft | Cars | DTD | Flowers | Pets | SUN397 | Caltech-101 | Food-101 | VOC2007 | Average |
|---|---|---|---|---|---|---|---|---|---|---|---|---|---|---|
| Real | SimCLR | | 88.3 | 70.3 | 47.1 | 45.5 | 76.2 | 92.5 | 70.1 | 65.4 | 83.8 | 75.0 | 81.2 | 72.3 |
| | SimCLR | ✓ | 92.3 | 75.4 | 47.8 | 44.4 | 77.1 | 91.8 | 69.2 | 65.1 | 84.7 | 74.9 | 81.2 | 73.1 (+0.8) |
| | CLIP | | 94.0 | 79.0 | 53.2 | 75.8 | 75.7 | 96.0 | 86.7 | 72.5 | 92.7 | 81.6 | 86.1 | 81.2 |
| | CLIP | ✓ | 95.8 | 82.9 | 51.5 | 76.5 | 74.7 | 95.3 | 87.2 | 72.5 | 92.7 | 81.7 | 86.2 | 81.5 (+0.3) |
| Syn | SimCLR | | 84.8 | 65.2 | 51.0 | 53.2 | 74.5 | 93.3 | 74.2 | 65.0 | 81.7 | 74.8 | 81.8 | 72.7 |
| | SimCLR | ✓ | 92.6 | 76.2 | 50.4 | 53.2 | 74.7 | 93.3 | 74.7 | 64.8 | 86.0 | 74.7 | 81.2 | 74.7 (+2.0) |
| | CLIP | | 87.3 | 69.5 | 53.5 | 79.5 | 75.8 | 95.4 | 85.8 | 69.2 | 90.9 | 78.3 | 84.5 | 79.1 |
| | CLIP | ✓ | 93.4 | 78.8 | 52.4 | 79.6 | 75.1 | 95.0 | 85.0 | 69.5 | 90.9 | 78.4 | 84.7 | 80.3 (+1.2) |
| | StableRep | | 90.7 | 74.4 | 57.6 | 80.3 | 79.0 | 96.7 | 87.1 | 73.2 | 94.0 | 83.5 | 87.2 | 82.2 |
| | StableRep | ✓ | 96.2 | 84.1 | 58.3 | 80.9 | 78.1 | 97.2 | 87.5 | 73.0 | 94.6 | 83.6 | 87.2 | 83.7 (+1.5) |

Table 8: We ablate the effects of `Random Downsample` augmentation on linear probing benchmarks from various domains. This augmentation brings significant improvements on CIFAR-10 and CIFAR-100 datasets, while maintaining the performance on other datasets.

## A.2  Standard self-supervised learning

We follow the default settings for standard self-supervised learning algorithms, and present the training details in Table 9 and Table 10. We use the linear $lr$ scaling rule: $lr = base\_lr \times bsz/256$. For BYOL [25], we did not follow the hyperparameters ($blr = 1.0e-4, wd = 0.03$) in [11], as we found our setting here yielded better accuracy. For DINO [6], we did not use the multi-crop strategy and only pre-trained the model with two 224×224 crops.

| config | MAE | SimCLR |
|---|---|---|
| optimizer | AdamW | AdamW |
| base learning rate | 1.5e-4 | 2.0e-4 |
| weight decay | 0.05 | 0.1 |
| optimizer momentum | $\beta_1, \beta_2 = 0.9, 0.95$ | $\beta_1, \beta_2 = 0.9, 0.98$ |
| batch size | 4096 | 4096 |
| learning rate schedule | cosine decay | cosine decay |
| epochs | 300 (cc3m) / 80 (cc12m) | 100 (cc3m) / 35 (cc12m) |
| warmup epochs | 10 (cc3m) / 4 (cc12m) | 5 (cc3m) / 1 (cc12m) |
| augmentation | RandomResizedCrop, Flip | SimCLR Aug. [10] |

Table 9: **Self-supervised pre-training settings.** MAE and SimCLR.

| config | DINO | BYOL/MoCo-v3 |
|---|---|---|
| optimizer | AdamW | AdamW |
| base learning rate | 5.0e-4 | 1.5e-4 |
| weight decay | 0.04 to 0.4, cosine | 0.1 |
| optimizer momentum | $\beta_1, \beta_2 = 0.9, 0.999$ | $\beta_1, \beta_2 = 0.9, 0.95$ |
| batch size | 4096 | 4096 |
| learning rate schedule | cosine decay | cosine decay |
| epochs | 100 (cc3m) / 35 (cc12m) | 100 (cc3m) / 35 (cc12m) |
| warmup epochs | 5 (cc3m) / 2 (cc12m) | 5 (cc3m) / 2 (cc12m) |
| momentum update $\lambda$ | 0.996 to 1, cosine | 0.996 to 1, cosine |
| augmentation | BYOL Aug. [25] | BYOL Aug. [25] |
| teacher temp. $\tau_t$ | 0.04 to 0.07 in warmup | |
| student temp. $\tau_s$ | 0.1 | |

Table 10: **Self-supervised pre-training settings.** DINO, BYOL and MoCo v3.

## A.3 StableRep pre-training

The hyperparameterss for StableRep is presented in Table 11. Indeed, they are the same as that in SimCLR. The difference is that the $base\_lr$ in StableRep is for 512 images while in SimCLR it is for 256 images, because each image in StableRep only has one single crop. We ended up using a batch size of 8256 images, since we trained our model with 32 GPUs and 8192 is not divisible over $32 \times 6$. The computation for StableRep has been converted to SimCLR-equivalent epochs.

| config | StableRep |
|---|---|
| batch size | 8256 ($m = 6$, $n = 1376$) |
| optimizer | AdamW |
| base learning rate | 2.0e-4 |
| peak learning rate | $base\_lr \times bsz/512$ |
| weight decay | 0.1 |
| optimizer momentum | $\beta_1, \beta_2 = 0.9, 0.98$ |
| learning rate schedule | cosine decay |
| epochs | 35 / 70 / 105 |
| warmup epochs | 1.2 / 2.3 / 3.5 |
| augmentation | Downsample Aug. + SimCLR Aug. [10] |

Table 11: **StableRep pre-training settings.**

## A.4 CLIP training

| config | CLIP |
|---|---|
| batch size | 8192 |
| optimizer | AdamW |
| peak learning rate | 1e-3 |
| weight decay | 0.5 |
| optimizer momentum | $\beta_1, \beta_2 = 0.9, 0.98$ |
| learning rate schedule | cosine decay |
| epochs | 35 |
| warmup epochs | 1 |
| augmentation | RandomResizedCrop(scale=(0.5, 1.0)) |

Table 12: **CLIP training settings.**

| Model | Patch size | Input resolution | Embedding dimension | Vision Transformer | | | Text Transformer | | | Vocab size | Text length |
|---|---|---|---|---|---|---|---|---|---|---|---|
| | | | | Layers | Width | Heads | Layers | Width | Heads | | |
| ViT-B/16 | 16 | 224 | 512 | 12 | 768 | 12 | 12 | 512 | 8 | 49,408 | 77 |

Table 13: **CLIP encoder details.**

We follow the hyperparameter setting used in [51] since it is better than that from the original CLIP [58] paper. Table 12 summarizes the training details, and Table 13 presents the architecture of CLIP encoders. With this training setup, we are able to produce $40.2\%$ ImageNet zero-shot accuracy when training CLIP on CC12M dataset. As a comparison, [51] reports $36.0\%$ using the same architecutre.

## A.5 ImageNet linear probing

We follow prior work [11, 6] to train the linear classifier. It has been generally observed that regularization such as weight decay hurts the performance [78]. Following [78, 11], we set weight decay as 0, and only use `RandomResizedCrop` and `RandomHorizontalFlip` as data augmentation. We sweep the $base\_lr$ over $\{0.1, 0.2, 0.5, 1, 2, 5, 10, 20, 50\} \times 10^{-2}$.

| config | value |
|---|---|
| batch size | 1024 |
| optimizer | SGD |
| base learning rate | sweep |
| weight decay | 0 |
| optimizer momentum | 0.9 |
| learning rate schedule | cosine decay |
| epochs | 90 |
| augmentation | RandomResizedCrop, Flip |

Table 14: **ImageNet linear probing settings.**

For StableRep trained with 35 epochs, we find that adding an extra BatchNorm layer without affine transformation improves and stablizes the linear probing results. However, this additional BatchNorm does not help when StableRep is trained with a longer schedule, e.g., 105 epochs. We conjecture that BatchNorm is helpful when StableRep is not convergent, and present the comparison in Table 15.

| | w/ BN | w/o BN |
|---|---|---|
| StableRep, 35 epochs | 73.5 | 71.4 |
| StableRep, 105 epochs | 75.2 | 75.4 |

Table 15: ImageNet linear probing results w/ or w/o extra BatchNorm layer for the linear classifier.

## A.6 Fine-grained linear classification

Following [10, 25, 20], we fit a regularized multinomial logistic regression model on top of the frozen `CLS` token. In training and testing, we do not perform any data augmentation; images are resized to 224 pixels along the shorter side using bicubic resampling, followed by a center crop of 224×224. We minimize the cross-entropy objective using L-BFGS with $\ell_2$-regularization. We select this $\ell_2$-regularization constant on the validation set over 45 logarithmically spaced values between $10^{-6}$ and $10^5$. The maximum number of L-BFGS iterations is set to $500$.

The details about the fine-grained classification datasets are presented in Table 16.

## A.7 Few-shot image classification

Following the settings in [20, 19], we evaluate the 5-way 5-shot performance on 10 different datasets. We do not use data augmentation; images are resized to 224 pixels along the shorter side using

| Dataset | Metric | Categories | Train Size | Test Size |
|---|---|---|---|---|
| CIFAR-10 [42] | Accuracy | 10 | 50,000 | 10,000 |
| CIFAR-100 [42] | Accuracy | 100 | 50,000 | 10,000 |
| Aircraft [47] | Mean per class | 100 | 6,667 | 3,333 |
| Cars [41] | Accuracy | 196 | 8,144 | 8,041 |
| DTD [13] | Accuracy | 47 | 3,760 | 1,880 |
| Flowers [53] | Mean per class | 102 | 2,040 | 6,149 |
| Pets [55] | Mean per class | 37 | 3,680 | 3,669 |
| SUN397 [85] | Accuracy | 397 | 19,850 | 19,850 |
| Caltech-101 [22] | Mean per class | 102 | 3,060 | 6,085 |
| Food-101 [5] | Accuracy | 101 | 75,750 | 25,250 |
| VOC2007 [21] | Mean per class | 20 | 5,011 | 4,952 |

Table 16: Details of the fine-grained linear classification datasets.

bicubic resampling, followed by a center crop of 224×224. We report the mean accuracy of 600 randomly sampled tasks (also known as episodes). For each task, images are randomly sampled from the combination of training, validation and testing sets. We sample 15 query images for each class in every task for evaluation purpose.

# B  Additional Results

## B.1  Fine-grained classification

| | | CIFAR-10 | CIFAR-100 | Aircraft | Cars | DTD | Flowers | Pets | SUN397 | Caltech-101 | Food-101 | VOC2007 | Average |
|---|---|---|---|---|---|---|---|---|---|---|---|---|---|
| | \multicolumn{12}{c}{Pre-training on Redcaps for 35 epochs} | | | | | | | | | | | |
| Real | SimCLR | 90.2 | 72.0 | 46.8 | 42.8 | 77.9 | 94.6 | 83.0 | 61.2 | 82.7 | 81.3 | 80.9 | 73.9 |
| Real | CLIP | 94.2 | 78.9 | 52.9 | 74.9 | 73.9 | 97.8 | **91.6** | 66.2 | 91.6 | **89.2** | 85.4 | 81.5 |
| Syn | SimCLR | 85.1 | 65.4 | 48.7 | 53.7 | 74.6 | 95.0 | 79.6 | 61.8 | 84.5 | 79.7 | 80.4 | 73.5 |
| Syn | CLIP | 88.7 | 71.4 | 53.7 | 77.3 | 76.0 | 96.9 | 88.2 | 67.3 | 90.3 | 83.7 | 84.5 | 79.8 |
| Syn | StableRep | **96.7** | **84.6** | **57.2** | **78.8** | **79.0** | **98.4** | 90.9 | **70.7** | **94.9** | 88.1 | **86.6** | **84.2** |
| | \multicolumn{12}{c}{Longer training for StableRep} | | | | | | | | | | | |
| cc12m | 35 epochs | 96.2 | 84.1 | 58.3 | 80.9 | 78.1 | 97.2 | 87.5 | 73.0 | 94.6 | 83.6 | 87.2 | 83.7 |
| cc12m | 105 epochs | 96.7 | 84.7 | 59.2 | 83.5 | 80.1 | 97.3 | 88.3 | 74.3 | 94.7 | 85.1 | 87.9 | 84.7 |
| redcaps | 35 epochs | 96.7 | 84.6 | 57.2 | 78.8 | 79.0 | 98.4 | 90.9 | 70.7 | 94.9 | 88.1 | 86.6 | 84.2 |
| redcaps | 105 epochs | 96.9 | 85.6 | 60.2 | 83.9 | 80.0 | 98.5 | 91.7 | 72.5 | 94.8 | 89.4 | 87.8 | 85.6 |
| | \multicolumn{12}{c}{OpenAI's CLIP trained on WIT-400M (numbers copied from [58])} | | | | | | | | | | | |
| | WIT-400M | 96.2 | 83.1 | 59.5 | 86.7 | 79.2 | 98.1 | 93.1 | 78.4 | 94.7 | 92.8 | 89.2 | 86.5 |

Table 17: Linear transfer results on fine-grained datasets. All results are with ViT-B/16. Results of StableRep are  marked . **Upper:** different methods pre-trained on RedCaps. **Middle:** StableRep with different training schedules. **Lower:** OpenAI's CLIP trained on WIT-400M dataset. Our StableRep trained with *synthetic* images only is approaching the performance of CLIP trained with 400 millions of real images.

In Table 17, we further present the fine-grained linear classification results by models from RedCaps or models that are trained longer (2x or 3x longer). When pre-training on RedCaps, StableRep achieves the best average accuracy. Longer training of StableRep further improves transferability. Notably, our StableRep trained with *synthetic* images only is approaching the performance of OpenAI's CLIP trained with 400 millions of real images.

## B.2 Few-shot image classification

We further summarizes the few-shot image classification results in Table 18. The 95% confidence interval is provided. StableRep stands out on the majority of the evaluated datasets.

| | | CIFAR-10 | CIFAR-100 | Aircraft | Cars | DTD | Flowers | Pets | SUN397 | Caltech-101 | Food-101 | Average |
|---|---|---|---|---|---|---|---|---|---|---|---|---|
| | | Pre-training on cc12m | | | | | | | | | | |
| Real | SimCLR | $64.0_{\pm0.7}$ | $70.4_{\pm0.8}$ | $40.7_{\pm0.9}$ | $50.9_{\pm0.8}$ | $82.2_{\pm0.6}$ | $92.1_{\pm0.5}$ | $74.4_{\pm0.8}$ | $94.0_{\pm0.4}$ | $90.4_{\pm0.5}$ | $70.4_{\pm0.7}$ | 73.0 |
| | CLIP | $77.5_{\pm0.6}$ | $82.1_{\pm0.7}$ | $62.0_{\pm1.0}$ | $90.9_{\pm0.5}$ | $83.3_{\pm0.6}$ | $97.6_{\pm0.2}$ | $91.1_{\pm0.5}$ | $97.2_{\pm0.2}$ | $98.2_{\pm0.2}$ | $87.0_{\pm0.5}$ | 86.7 |
| Syn | SimCLR | $50.0_{\pm0.6}$ | $58.9_{\pm0.8}$ | $45.2_{\pm1.0}$ | $54.2_{\pm0.8}$ | $79.8_{\pm0.6}$ | $92.0_{\pm0.5}$ | $74.6_{\pm0.8}$ | $92.9_{\pm0.4}$ | $89.1_{\pm0.6}$ | $71.0_{\pm0.7}$ | 70.8 |
| | CLIP | $63.1_{\pm0.6}$ | $73.5_{\pm0.7}$ | $61.3_{\pm1.0}$ | $\mathbf{92.5_{\pm0.4}}$ | $81.7_{\pm0.6}$ | $96.9_{\pm0.3}$ | $91.5_{\pm0.5}$ | $96.7_{\pm0.2}$ | $96.8_{\pm0.3}$ | $82.5_{\pm0.6}$ | 83.7 |
| | StableRep | $\mathbf{92.3_{\pm0.3}}$ | $\mathbf{91.8_{\pm0.5}}$ | $\mathbf{62.6_{\pm1.0}}$ | $91.8_{\pm0.5}$ | $\mathbf{86.4_{\pm0.5}}$ | $\mathbf{98.2_{\pm0.2}}$ | $\mathbf{91.7_{\pm0.5}}$ | $\mathbf{97.3_{\pm0.2}}$ | $\mathbf{98.8_{\pm0.2}}$ | $\mathbf{87.3_{\pm0.5}}$ | 89.8 |
| | | Pre-training on redcaps | | | | | | | | | | |
| Real | SimCLR | $62.3_{\pm0.6}$ | $69.4_{\pm0.7}$ | $39.6_{\pm0.9}$ | $51.0_{\pm0.8}$ | $82.7_{\pm0.6}$ | $94.8_{\pm0.4}$ | $85.4_{\pm0.6}$ | $91.8_{\pm0.5}$ | $88.5_{\pm0.6}$ | $79.1_{\pm0.7}$ | 74.5 |
| | CLIP | $80.6_{\pm0.5}$ | $85.3_{\pm0.6}$ | $54.5_{\pm0.9}$ | $88.5_{\pm0.6}$ | $82.6_{\pm0.6}$ | $99.0_{\pm0.1}$ | $\mathbf{94.5_{\pm0.4}}$ | $95.9_{\pm0.3}$ | $97.8_{\pm0.2}$ | $\mathbf{94.4_{\pm0.3}}$ | 87.3 |
| Syn | SimCLR | $52.9_{\pm0.6}$ | $60.8_{\pm0.8}$ | $40.9_{\pm0.9}$ | $53.2_{\pm0.8}$ | $79.5_{\pm0.6}$ | $94.3_{\pm0.4}$ | $78.3_{\pm0.7}$ | $92.0_{\pm0.4}$ | $88.9_{\pm0.5}$ | $75.9_{\pm0.7}$ | 71.7 |
| | CLIP | $65.7_{\pm0.6}$ | $75.7_{\pm0.7}$ | $55.2_{\pm1.0}$ | $\mathbf{90.1_{\pm0.5}}$ | $82.6_{\pm0.6}$ | $98.2_{\pm0.2}$ | $92.0_{\pm0.5}$ | $96.3_{\pm0.3}$ | $96.9_{\pm0.3}$ | $88.1_{\pm0.5}$ | 84.1 |
| | StableRep | $\mathbf{92.7_{\pm0.3}}$ | $\mathbf{92.9_{\pm0.4}}$ | $\mathbf{57.3_{\pm1.0}}$ | $89.4_{\pm0.6}$ | $\mathbf{86.2_{\pm0.5}}$ | $\mathbf{99.2_{\pm0.1}}$ | $94.5_{\pm0.4}$ | $\mathbf{96.8_{\pm0.3}}$ | $\mathbf{98.9_{\pm0.2}}$ | $91.8_{\pm0.4}$ | 90.0 |

Table 18: Few-shot image classification results with 95% confidence interval provided. All models here are trained for 35 epochs. **Upper:** pre-training on CC12M dataset. **Upper:** pre-training on RedCaps dataset.

# C  Image Generation

## C.1  Implementation details

We use Stable Diffusion [61] v1.5. During sampling, we generate images by 50 DDIM [74] steps. To accelerate the generation process, we leverage xFormers library for efficient attention computation, which brings down the sampling time to ∼0.8s per image on a single A100 GPU and ∼2.3s per image on a V100 GPU.

**Image resolution.** The image resolution may affect the quality of representations learned by self-supervised learning algorithms. We try to make a relative fair comparison by storing all synthetic and real images in similar resolutions. The synthetic images generated by Stable Diffusion are 512×512; we resized them to 256×256 before storing them on the disk. The real images have various sizes, ranging from less than a hundred of pixels in shorter side to thousands of pixels; we resize the shorter side of all real images to 256.

## C.2  Generation examples

Some examples of synthetic images are visualized in Figure 8.

# D  Computation

**Synthesis.** The slowest part of the StableRep pipeline is the image generation. We use 512 V100 GPUs to synthesize images, which takes ∼13 hours for every ten million images.

**Pre-training.** Each of our StableRep models with ViT-B/16 is trained on 4 nodes, each of which has 8 A100 GPUs and 96 CPU cores. It takes ∼20 hours to complete 35 SimCLR-equivalent epochs of training on CC12M and ∼23 hours on RedCaps. For ViT-L/16, we use 64 A100 80GB GPUs spread over 8 nodes.

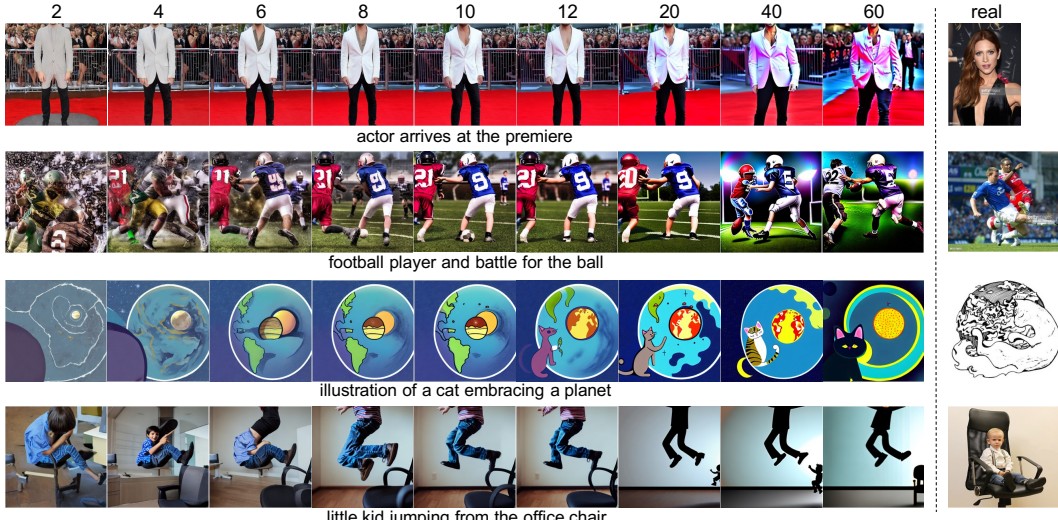

Figure 8: **Examples of synthetic images.** We show examples for 4 different text prompts. For each prompt, we provide examples synthesized with different guidance scale $w$, as well as the original real image.

