# A Implementation Details

## A.1 Standard self-supervised learning

We follow the default settings for standard self-supervised learning algorithms, and present the training details in Table 1 and Table 2. We use the linear $lr$ scaling rule: $lr = base\_lr \times bsz/256$. For **BYOL** [6], we did not follow the hyperparameters ($blr = 1.0e-4, wd = 0.03$) in [4], as we found our setting here yielded better accuracy. For **DINO** [2], we did not use the multi-crop strategy and only pre-trained the model with two 224×224 crops.

| config | MAE | SimCLR |
|---|---|---|
| optimizer | AdamW | AdamW |
| base learning rate | 1.5e-4 | 2.0e-4 |
| weight decay | 0.05 | 0.1 |
| optimizer momentum | $\beta_1, \beta_2{=}0.9, 0.95$ | $\beta_1, \beta_2{=}0.9, 0.98$ |
| batch size | 4096 | 4096 |
| learning rate schedule | cosine decay | cosine decay |
| epochs | 300 (cc3m) / 80 (cc12m) | 100 (cc3m) / 35 (cc12m) |
| warmup epochs | 10 (cc3m) / 4 (cc12m) | 5 (cc3m) / 1 (cc12m) |
| augmentation | RandomResizedCrop, Flip | SimCLR Aug. [3] |

Table 1: **Self-supervised pre-training settings.** MAE and SimCLR.

| config | DINO | BYOL/MoCo-v3 |
|---|---|---|
| optimizer | AdamW | AdamW |
| base learning rate | 5.0e-4 | 1.5e-4 |
| weight decay | 0.04 to 0.4, cosine schedule | 0.1 |
| optimizer momentum | $\beta_1, \beta_2{=}0.9, 0.999$ | $\beta_1, \beta_2{=}0.9, 0.95$ |
| batch size | 4096 | 4096 |
| learning rate schedule | cosine decay | cosine decay |
| epochs | 100 (cc3m) / 35 (cc12m) | 100 (cc3m) / 35 (cc12m) |
| warmup epochs | 5 (cc3m) / 2 (cc12m) | 5 (cc3m) / 2 (cc12m) |
| momentum update $\lambda$ | 0.996 to 1, cosine schedule | 0.996 to 1, cosine schedule |
| augmentation | BYOL Aug. [6] | BYOL Aug. [6] |
| teacher temp. $\tau_t$ | 0.04 to 0.07, linear schedule | |
| student temp. $\tau_s$ | 0.1 | |

Table 2: **Self-supervised pre-training settings.** DINO, BYOL and MoCo v3.

## A.2 StableRep pre-training

| config | StableRep |
|---|---|
| batch size | 8256 ($m = 6$, $n = 1376$) |
| optimizer | AdamW |
| base learning rate | 2.0e-4 |
| peak learning rate | $base\_lr \times bsz/512$ |
| weight decay | 0.1 |
| optimizer momentum | $\beta_1, \beta_2{=}0.9, 0.98$ |
| learning rate schedule | cosine decay |
| epochs | 35 / 70 / 105 |
| warmup epochs | 1.2 / 2.3 / 3.5 |
| augmentation | SimCLR Aug. [3] |

Table 3: **StableRep pre-training settings.**

The hyperparameterss for StableRep is presented in Table 3. Indeed, they are the same as that in SimCLR. The difference is that the $base\_lr$ in StableRep is for 512 images while in SimCLR it is for 256 images, because each image in StableRep only has one single crop. We ended up using a batch size of 8256 images, since we trained our model with 32 GPUs and 8192 is not divisible over $32\times6$. The computation for StableRep has been converted to SimCLR-equivalent epochs.

## A.3  CLIP training

We follow the hyperparameter setting used in [7] since it is better than that from the original CLIP [8] paper. Table 4 summarizes the training details, and Table 5 presents the architecture of CLIP encoders. With this training setup, we are able to produce $40.2\%$ ImageNet zero-shot accuracy when training CLIP on CC12M dataset. As a comparison, [7] reports $36.0\%$ using the same architecutre.

| config | CLIP |
|---|---|
| batch size | 8192 |
| optimizer | AdamW |
| peak learning rate | 1e-3 |
| weight decay | 0.5 |
| optimizer momentum | $\beta_1, \beta_2 = 0.9, 0.98$ |
| learning rate schedule | cosine decay |
| epochs | 35 |
| warmup epochs | 1 |
| augmentation | RandomResizedCrop(scale=(0.5, 1.0)) |

Table 4: **CLIP training settings.**

| Model | Patch size | Input resolution | Embedding dimension | Vision Transformer | | | Text Transformer | | | Vocab size | Text length |
|---|---|---|---|---|---|---|---|---|---|---|---|
| | | | | Layers | Width | Heads | Layers | Width | Heads | | |
| ViT-B/16 | 16 | 224 | 512 | 12 | 768 | 12 | 12 | 512 | 8 | 49,408 | 77 |

Table 5: **CLIP encoder details.**

## A.4  ImageNet linear probing

We follow prior work [4, 2] to train the linear classifier. It has been generally observed that regularization such as weight decay hurts the performance [11]. Following [11, 4], we set weight decay as 0, and only use `RandomResizedCrop` and `RandomHorizontalFlip` as data augmentation. We sweep the $base\_lr$ over $\{0.2, 0.5, 1, 1.5, 2, 3, 5, 10\} \times 10^{-2}$.

| config | value |
|---|---|
| batch size | 1024 |
| optimizer | SGD |
| base learning rate | sweep |
| weight decay | 0 |
| optimizer momentum | 0.9 |
| learning rate schedule | cosine decay |
| epochs | 90 |
| augmentation | RandomResizedCrop, Flip |

Table 6: **ImageNet linear probing settings.**

## A.5  Fine-grained linear classification

Following [3, 6, 5], we fit a regularized multinomial logistic regression model on top of the frozen `CLS` token. In training and testing, we do not perform any data augmentation; images are resized to 224 pixels along the shorter side using bicubic resampling, followed by a center crop of $224\times224$.

We minimize the cross-entropy objective using L-BFGS with $\ell_2$-regularization. We select this $\ell_2$-regularization constant on the validation set over 45 logarithmically spaced values between $10^{-6}$ and $10^5$. The maximum number of L-BFGS iterations is set to 500.

## A.6 Few-shot image classification

Following the settings in [5, 1], we evaluate the 5-way 5-shot performance on 10 different datasets. We do not use data augmentation; images are resized to 224 pixels along the shorter side using bicubic resampling, followed by a center crop of 224×224. We report the mean accuracy of 600 randomly sampled tasks (also known as episodes). For each task, images are randomly sampled from the combination of training, validation and testing sets. We sample 15 query images for each class in every task for evaluation purpose.

# B  Additional Results

## B.1  Fine-grained classification

In Table 7, we further present the fine-grained linear classification results by models from RedCaps or models that are trained longer (2x or 3x longer). When pre-training on RedCaps, StableRep achieves the best average accuracy. Longer training of StableRep further improves transferability.

| | | CIFAR-10 | CIFAR-100 | Aircraft | Cars | DTD | Flowers | Pets | SUN397 | Caltech-101 | Food-101 | VOC2007 | Average |
|---|---|---|---|---|---|---|---|---|---|---|---|---|---|
| | | | | | | Pre-training on Redcaps | | | | | | | |
| Real | SimCLR | 90.2 | 72.0 | 46.8 | 42.8 | 77.9 | 94.6 | 83.0 | 61.2 | 82.7 | 81.3 | 80.9 | 73.9 |
| | CLIP | **94.2** | **78.9** | 52.9 | 74.9 | 73.9 | 97.8 | 91.6 | 66.2 | **91.6** | **89.2** | 85.4 | 81.5 |
| Syn | SimCLR | 85.1 | 65.4 | 48.7 | 53.7 | 74.6 | 95.0 | 79.6 | 61.8 | 84.5 | 79.7 | 80.4 | 73.5 |
| | CLIP | 88.7 | 71.4 | 53.7 | 77.3 | 76.0 | 96.9 | 88.2 | 67.3 | 90.3 | 83.7 | 84.5 | 79.8 |
| | StableRep | 90.4 | 73.8 | **57.5** | **81.1** | **79.5** | **98.4** | 90.8 | **71.1** | **95.1** | 88.2 | **86.7** | **83.0** |
| | | | | | | Longer training for StableRep | | | | | | | |
| cc12m | 35 epochs | 90.7 | 74.4 | 57.6 | 80.3 | 79.0 | 96.7 | 87.1 | 73.2 | 94.0 | 83.5 | 87.2 | 82.2 |
| | 70 epochs | 91.5 | 74.7 | 59.1 | 82.5 | 79.7 | 97.5 | 88.1 | 74.3 | 94.3 | 85.0 | 87.8 | 83.1 |
| | 105 epochs | 91.5 | 75.9 | 58.8 | 84.2 | 80.1 | 97.6 | 87.9 | 74.7 | 94.5 | 85.4 | 87.8 | **83.5** |
| redcaps | 35 epochs | 90.4 | 73.8 | 57.5 | 81.1 | 79.5 | 98.4 | 90.8 | 71.1 | 95.1 | 88.2 | 86.7 | 83.0 |
| | 70 epochs | 91.0 | 75.4 | 58.0 | 83.3 | 79.8 | 98.5 | 90.7 | 72.9 | 95.2 | 89.3 | 87.5 | 83.8 |
| | 105 epochs | 91.3 | 75.0 | 59.6 | 82.8 | 80.7 | 98.6 | 91.0 | 72.8 | 94.7 | 89.2 | 87.6 | **83.9** |

Table 7: Linear transfer results on fine-grained datasets. **Upper:** different methods pre-trained on RedCaps. **Lower:** StableRep with different training schedules on CC12M and RedCaps. Longer training improves transferability.

## B.2  Few-shot image classification

We further summarizes the few-shot image classification results in Table 8. The 95% confidence interval is provided. StableRep stands out on the majority of the evaluated datasets.

# C  Image Generation

## C.1  Implementation details

We use Stable Diffusion [9] v1.5. During sampling, we generate images by 50 DDIM [10] steps. To accelerate the generation process, we leverage xFormers library for efficient attention computation, which brings down the sampling time to ∼0.8s per image on a single A100 GPU and ∼2.3s per image

| | | CIFAR-10 | CIFAR-100 | Aircraft | Cars | DTD | Flowers | Pets | SUN397 | Caltech-101 | Food-101 | **Average** |
|---|---|---|---|---|---|---|---|---|---|---|---|---|
| | | | | | | Pre-training on cc12m | | | | | | |
| Real | SimCLR | $64.0_{\pm0.7}$ | $70.4_{\pm0.8}$ | $40.7_{\pm0.9}$ | $50.9_{\pm0.8}$ | $82.2_{\pm0.6}$ | $92.1_{\pm0.5}$ | $74.4_{\pm0.8}$ | $94.0_{\pm0.4}$ | $90.4_{\pm0.5}$ | $70.4_{\pm0.7}$ | 73.0 |
| Real | CLIP | $\mathbf{77.5_{\pm0.6}}$ | $\mathbf{82.1_{\pm0.7}}$ | $62.0_{\pm1.0}$ | $90.9_{\pm0.5}$ | $83.3_{\pm0.6}$ | $97.6_{\pm0.2}$ | $91.1_{\pm0.5}$ | $97.2_{\pm0.2}$ | $98.2_{\pm0.2}$ | $87.0_{\pm0.5}$ | 86.7 |
| Syn | SimCLR | $50.0_{\pm0.6}$ | $58.9_{\pm0.8}$ | $45.2_{\pm1.0}$ | $54.2_{\pm0.8}$ | $79.8_{\pm0.6}$ | $92.0_{\pm0.5}$ | $74.6_{\pm0.6}$ | $92.9_{\pm0.6}$ | $89.1_{\pm0.6}$ | $71.0_{\pm0.7}$ | 70.8 |
| Syn | CLIP | $63.1_{\pm0.6}$ | $73.5_{\pm0.7}$ | $61.3_{\pm1.0}$ | $\mathbf{92.5_{\pm0.4}}$ | $81.7_{\pm0.6}$ | $96.9_{\pm0.3}$ | $91.5_{\pm0.5}$ | $96.7_{\pm0.2}$ | $96.8_{\pm0.3}$ | $82.5_{\pm0.6}$ | 83.7 |
| Syn | StableRep | $68.2_{\pm0.6}$ | $75.9_{\pm0.8}$ | $\mathbf{62.5_{\pm1.0}}$ | $92.0_{\pm0.5}$ | $\mathbf{86.3_{\pm0.5}}$ | $\mathbf{98.2_{\pm0.2}}$ | $\mathbf{92.4_{\pm0.5}}$ | $\mathbf{97.3_{\pm0.2}}$ | $\mathbf{98.7_{\pm0.2}}$ | $\mathbf{87.6_{\pm0.5}}$ | 85.9 |
| | | | | | | Pre-training on redcaps | | | | | | |
| Real | SimCLR | $62.3_{\pm0.6}$ | $69.4_{\pm0.7}$ | $39.6_{\pm0.9}$ | $51.0_{\pm0.8}$ | $82.7_{\pm0.6}$ | $94.8_{\pm0.4}$ | $85.4_{\pm0.6}$ | $91.8_{\pm0.5}$ | $88.5_{\pm0.6}$ | $79.1_{\pm0.7}$ | 74.5 |
| Real | CLIP | $\mathbf{80.6_{\pm0.5}}$ | $\mathbf{85.3_{\pm0.6}}$ | $54.5_{\pm0.9}$ | $88.5_{\pm0.6}$ | $82.6_{\pm0.6}$ | $99.0_{\pm0.1}$ | $\mathbf{94.5_{\pm0.4}}$ | $95.9_{\pm0.3}$ | $97.8_{\pm0.2}$ | $\mathbf{94.4_{\pm0.3}}$ | 87.3 |
| Syn | SimCLR | $52.9_{\pm0.6}$ | $60.8_{\pm0.8}$ | $40.9_{\pm0.9}$ | $53.2_{\pm0.8}$ | $79.5_{\pm0.6}$ | $94.3_{\pm0.4}$ | $78.3_{\pm0.7}$ | $92.0_{\pm0.4}$ | $88.9_{\pm0.5}$ | $75.9_{\pm0.7}$ | 71.7 |
| Syn | CLIP | $65.7_{\pm0.6}$ | $75.7_{\pm0.7}$ | $55.2_{\pm1.0}$ | $\mathbf{90.1_{\pm0.5}}$ | $82.6_{\pm0.6}$ | $98.2_{\pm0.2}$ | $92.0_{\pm0.5}$ | $96.3_{\pm0.3}$ | $96.9_{\pm0.3}$ | $88.1_{\pm0.5}$ | 84.1 |
| Syn | StableRep | $68.0_{\pm0.6}$ | $76.7_{\pm0.8}$ | $\mathbf{57.1_{\pm1.0}}$ | $\mathbf{90.1_{\pm0.5}}$ | $\mathbf{86.5_{\pm0.5}}$ | $\mathbf{99.2_{\pm0.1}}$ | $94.4_{\pm0.4}$ | $\mathbf{96.9_{\pm0.3}}$ | $\mathbf{98.9_{\pm0.2}}$ | $92.1_{\pm0.4}$ | 86.0 |

Table 8: Few-shot image classification. **Upper:** pre-training on CC12M dataset. **Upper:** pre-training on RedCaps dataset.

on a V100 GPU. We use 512 V100 GPUs to synthesize images in large scale, which takes ∼13 hours for every ten million images.

**Image resolution.** The image resolution may affect the quality of representations learned by self-supervised learning algorithms. We try to make a relative fair comparison by storing all synthetic and real images in similar resolutions. The synthetic images generated by Stable Diffusion are 512×512; we resized them to 256×256 before storing them on the disk. The real images have various sizes, ranging from less than a hundred of pixels in shorter side to thousands of pixels; we resize the shorter side of all real images to 256.

## C.2 Generation Examples

Some examples of synthetic images are visualized in Figure 1.

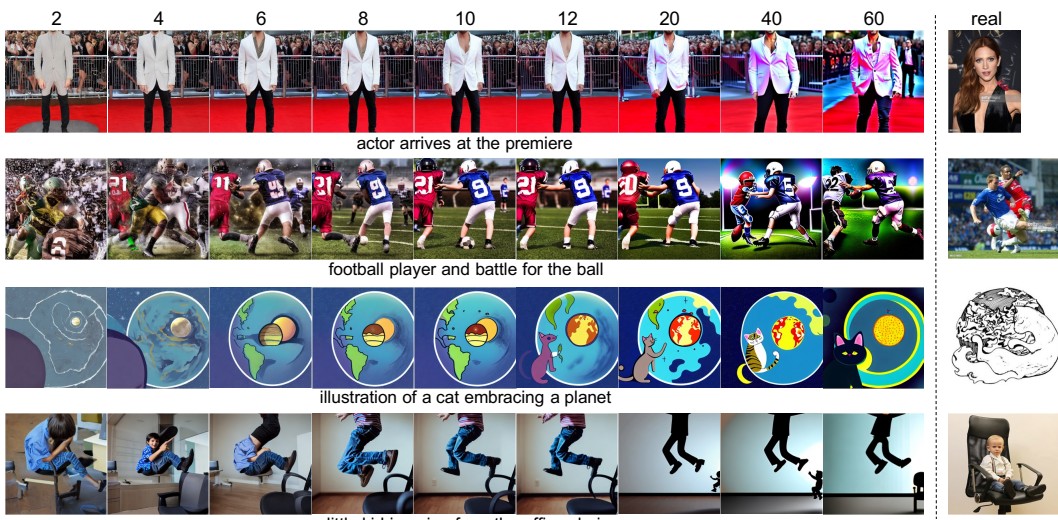

actor arrives at the premiere

football player and battle for the ball

illustration of a cat embracing a planet

little kid jumping from the office chair

Figure 1: **Examples of synthetic images.** We show examples for 4 different text prompts. For each prompt, we provide examples synthesized with different guidance scale $w$, as well as the original real image.

# D   Further Discussion

**Broader impacts.** This paper is on the basics of visual representation learning, and we believe it will be beneficial to the practice of this field. An immediate application of our method is to reduce the reliance on collecting large scale real images for learning representations. This may have the beneficial effects of being more cost effective and reducing biases introduced by human collection and curation processes. At the same time, our method relies on pre-trained text-to-image generative models that are trained on large scale uncurated web-scale data, and such data may hide social biases and errors that would have been uncovered via the human curation process. We also note that the text prompts we used are not bias free: what prompts we choose determine what images are synthesized. The choice of prompts therefore plays a similar role to the choice of what real images to collect for self-supervised visual representation learning.

**Compute.** Each of our StableRep models is trained on 4 nodes, each of which has 8 A100 GPUs and 96 CPU cores. We store all synthetic images inside two NFS folders, each with 100 TBs. It takes $\sim$20 hours to complete 35 SimCLR-equivalent epochs of training on CC12M and $\sim$23 hours on RedCaps.