# OpenReview forum: "StableRep: Synthetic Images from Text-to-Image Models Make Strong Visual Representation Learners"
_NeurIPS.cc/2023/Conference — NeurIPS 2023 poster_

### Official Review · Reviewer_FSjQ · 2023-06-10

**Soundness:** 3 good
**Presentation:** 3 good
**Contribution:** 4 excellent
**Rating:** 8
**Confidence:** 4

**Summary:**

The paper investigates the potential of using synthetic images generated by text-to-image models to train self-supervised image embedding models. Methodologically, the authors propose StableRep, a multi-positive contrastive learning method that treats multiple images generated from the same text prompt as positive examples for each other. The paper demonstrates two key findings. First, when the generative model is appropriately configured, self-supervised methods trained on synthetic images can achieve comparable or superior performance to real image counterparts. Secondly, when language supervision is incorporated, synthetic data become more efficient.

**Strengths:**

1. **Interesting topic:** The paper investigates the potential of using synthetic images generated by text-to-image models as strong visual representation learners. This exploration addresses an important and timely question.

2. **New contrastive learning method:** The proposed StableRep is new and tailored for synthetic data.

3. **Comprehensive evaluation:** The paper evaluates the representations learned by StableRep on large-scale datasets and compares them with strong baselines.

**Weaknesses:**

1. **Potentially entangled comparison in terms of the amount of data:** The only concern I have for this paper is the claim of data amount. When claiming "synthetic and real images of the same amount", the authors refer to the amount of data for representation learning. However, Stable Diffusion is trained on LAION-2B. Therefore, 50M synthetic images may contain information from 2B images, making this comparison less meaningful. I'd like to see more discussion on this issue.

**Questions:**

See Weaknesses.

**Limitations:**

Limitations are included in Section 7.

---

> ### Author Rebuttal · Authors · 2023-08-10
>
> Thank you very much for considering our work as interesting and novel! The concern you raised is a great question, and we will try to answer it from two aspects:
>
>  1. While the dataset distillation hypothesis is unavoidable, we have a different interpretation from the "distribution" perspective. Suppose there is an underlying distribution of all natural images, e.g., every single scene on this planet projected into an image via a pinhole camera. We call such underlying distribution $D$.  LAION-2B is just a set of 2B images sampled from $D$, and we used LAION-2B to estimate this underlying distribution and then yielded a stable diffusion model, which can be viewed as a "parameterized distribution" $D_{sd}$ that approximate $D$. In theory, $D$ is still much more powerful than $D_{sd}$. Then if we are about to sample some random training images fro representation learning from either $D$ or $D_{sd}$, which one should I pick? One may choose $D$ because it's theoretically more powerful, but we argue $D_{sd}$ may be a good (or even better) option, why? Because $D_{sd}$ allows us to have more fine-grained control over how we sample those images: (1) guidance scale allows us to trade off between image quality and diversity so we can tune it, but for $D$ it's very expensive or even infeasible to tune it -- once the data is collected it's collected; (2) the stochasticity of diffusion models allows us to sample multiple images that correspond to the same underlying semantics described by an caption, but doing so with $D$ is almost infeasible; (3) and potentially others. So in summary, $D_{sd}$ is a weaker distribution and randomly sampling from it may actually be disadvantageous, rather than than advantageous implied by the distillation hypothesis. But through the control of (1) and (2), we can sample from $D_{sd}$ in a more efficient way than $D$ allows us to do.
>
> 2. Practically, we conduct an surrogate sanity check to get a rough sense of how much information the synthesized training data contains about the downstream task. Specifically, we compute how close the downstream datasets are to the StableRep's synthetic training set. Concretely, for an image $I$ in a downstream dataset, we find its closest image in StableRep's training set by using cosine similarity of some pre-trained SimCLR feature extractor $f$ (Recall that SimCLR is optimized to maximize cosine similarity of the same image). We define this highest similarity (noted as $s$) as this image's similarity with StableRep's training set $D$. Concretely, $s_i = \max_{X \in D} cos(f(I_i), f(X))$. We evaluate four datasets, and provide the statistics (avg, min, and max) of this highest similarity metric of each image.
>
> | Train Set | stats | aircraft|cars|flowers|food101|
> |--|:--:|:--:|:--:|:--:|:--:|
> | StableRep Train | avg | 0.891 | 0.836 | 0.836 | 0.791 |
> |                 | min | 0.652 | 0.258 | 0.523 | 0.443 |
> |                 | max | 0.985 | 0.980 | 0.967 | 0.958 |
> | ImageNet  Train | avg | 0.901 | 0.870 | 0.823 | 0.807 |
> |                 | min | 0.700 | 0.629 | 0.586 | 0.560 |
> |                 | max | 0.980 | 0.998 | 0.988 | 0.977 |
>
> As a comparison, we also include the distance to ImageNet training set as a baseline. We observe that:
> - generally downstream datasets are closer (by avg metric) to ImageNet training set than our StableRep train set, except for flowers.
> - there is no value surpass 0.99 for StableRep train set, while there is 0.998 for ImageNet train set.
>
> From this perspective, we may conjecture that the ImageNet train set contains more information about the downstream dataset than the StableRep dataset. However, the StableRep still generalizes better than methods trained on real ImageNet train set, as illustrated in question 1 of the "global rebuttal" shared to all viewers.

---

> > ### Comment · Reviewer_FSjQ · 2023-08-17
> >
> > Thank you for your response. The first argument can be further strengthened by an experimental setting where collecting 50M high-quality images is almost infeasible for $D$ but easy for $D_{sd}$, but the proof-of-concept experiment in the paper is interesting enough to me. Overall, I think this paper is very interesting to the generative model and representation learning communities, and I'd like to raise my score to 8.

---

> > > ### Author Response · Authors · 2023-08-19
> > > **Thank you and some update**
> > >
> > > Dear Reviewer FSjQ,
> > >
> > > Thank you for reading our arguments and proposing this new setting!
> > >
> > > We have also obtained some new results after the rebuttal deadline, which may be interesting to you. So we want to share with you a summarization of all additional results during this period, which can be found [here](https://openreview.net/forum?id=xpjsOQtKqx&noteId=F3K5WcycnW).
> > >
> > > We would be very happy if they can address any of your remained concerns, or enhance your evaluation about our work, or simply just let you feel our work is more interesting. If you have any unsolved/new concerns, feedbacks, or comments stimulated by these additional results, please do let us know.

---

### Official Review · Reviewer_DyTC · 2023-06-29

**Soundness:** 3 good
**Presentation:** 3 good
**Contribution:** 3 good
**Rating:** 7
**Confidence:** 5

**Summary:**

In this paper, the authors investigate the potential of learning visual representations using synthetic data generated by text-to-image models. The authors choose Stable Diffusion for exploration and extensive experiments demonstrate that self-supervised models trained on synthetic data can perform better or at par with training on real data. The authors also propose a multi-positive contrastive variant called StableRep to allow multiple images generated from the same text prompt as positives for each other.

**Strengths:**

- The paper is well motivated. Given the recent progress of large-scale text-to-image generative models like Stable Diffusion, it is timing to investigate the effectiveness of these models in generating high-quality images to assist discriminative tasks.

- The paper is generally well-written and easy to follow.

- The experiments are extensive and the results seem promising.

**Weaknesses:**

- The authors use large-scale image-text datasets like CC3M, CC12M, RedCaps for study. These datasets have well-collected image captions. Although the authors claim that the proposed method can reduce the reliance on collecting large-scale real images for learning representations, the cost of collecting these image captions is also somewhat expensive and should not be ignored. Thus, a more cost-effective and interesting setting should be to generate a synthetic ImageNet dataset using category labels as text prompts (e.g., a photo of [category]) since it does not require any human effects of collecting captions. Then self-supervised models could be pre-trained on the real ImageNet dataset as well as the synthetic ImageNet dataset for comparisons. I am curious about whether the self-supervised methods trained on synthetic ImageNet still have advantages over training on real ImageNet.

- It seems that the synthetic data only have performance advantages for linear probing experiments. For few-shot experiments, self-supervised methods trained on synthetic data still have a large performance gap with training on real data, which limits the practical applications of using synthetic data to some extent.

- The authors only evaluate the proposed method on classification downstream tasks. What about dense prediction downstream tasks like object detection or semantic segmentation? Will the synthetic data still have the advantages on these kinds of downstream tasks?

- According to ablations in Sec. 4.2, generating multiple images per caption yields better performance. However, this may be due to simply increasing the number of views. To eliminate the interference of the number of views, the authors should also conduct an ablation by replacing multiple generated images per caption with the same number of cropped views per real image.

**Questions:**

I am concerned about the questions mentioned above. Given the current status of the paper, I am leaning towards borderline accept and hope the authors could address my concerns.

**Limitations:**

The authors have addressed the limitations and broader impacts in Sec. 7 (main text) and Sec. D (supplementary material) in detail, which look good to me.

---

> ### Author Rebuttal · Authors · 2023-08-10
>
> Thank you for providing valuable feedback. Below we try to address your concerns:
>
> > Thus, a more cost-effective and interesting setting should be to generate a synthetic ImageNet dataset using category labels as text prompts (e.g., a photo of [category])...I am curious about whether the self-supervised methods trained on synthetic ImageNet still have advantages over training on real ImageNet.
>
> This is a good suggestion. We have conducted the experiments and include the results in question 1 of the "global rebuttal" shared to all reviewers (it is located around the top of this page).
>
> In terms of Synthetic ImageNet v.s. Real ImageNet, we observe that:
>
> - When conduct supervised cross-entropy training, Real data significantly outperforms Synthetic data on either downstream linear classification or few-shot evaluation.
> - When conduct self-supervised learning, i.e., SimCLR, Synthetic ImageNet is comparable to (or just slightly worse than) Real ImageNet on downstream linear probing benchmarks. Generally, different prompting strategies result in different performance, and this indicates the importance of designing prompts.
>
> We also observe that, StableRep consistently outperforms both supervised training and SimCLR with synthetic ImageNet, demonstrating the effectiveness of our pipeline. While collecting image captions can be more expensive than using labels, we argue that we can explore large language models to help us synthesize these captions, which may becoming increasingly accessible and cheaper in the near future.
>
> > It seems that the synthetic data only have performance advantages for linear probing experiments. For few-shot experiments, self-supervised methods trained on synthetic data still have a large performance gap with training on real data, which limits the practical applications of using synthetic data to some extent.
>
> This is not completely true.
>
> Firstly, our StableRep outperforms training with real data on 8 out of the 10 few-shot benchmarks, and the average accuracy (over all 10 datasets) only lags behind by 0.8%.
>
> Secondly, the impression of underperfomance (for synthetic training) indeed comes from cifar-10/100. We have identified this as image resolution issues (please check our experiments in question 2 of the "global rebuttal"). Intuitively, the real training images contain low-resolution and blurring images, which help the model generalize to extremely low resolutions, such as 32x32 for cifar. In contrast, the synthetic data only contains high resolution images. A potential fix could be randomly downsampling images when training models on synthetic data.
>
> > The authors only evaluate the proposed method on classification downstream tasks. What about dense prediction downstream tasks like object detection or semantic segmentation?
>
> This is a great suggestion,  and we have evaluated the quality of representation on ADE20k semantic segmentation dataset.  Specifically, we freeze the backbone, and train the decoder part of an UperNet [a]. We used the default parameters in the mmsegmentation library [b]. The results are as below:
>
> | method | pre-train data | mean IoU (%) | pixel acc. (%)|
> |--|:--:|:--:|:--:|
> | clip | real, cc12m | 33.3 | 75.9 |
> | simclr | real, cc12m | 35.2 | 77.6 |
> | simclr | syn, cc12m  | 33.3 | 76.0 |
> | StableRep | syn, cc12m | **38.0** | **78.4** |
> | StableRep | syn, RedCaps | 37.1 | 77.8 |
>
> We observe that our StableRep consistently outperforms all other methods.
>
>
> > According to ablations in Sec. 4.2, generating multiple images per caption yields better performance. However, this may be due to simply increasing the number of views. To eliminate the interference of the number of views, the authors should also conduct an ablation by replacing multiple generated images per caption with the same number of cropped views per real image.
>
> This is a good suggestion for ablation study. To keep the comparison with StableRep sensible, we keep the total amount of crops per-batch as 8192, and vary the number of crops per image. In other words, if we use k views per image for SimCLR, then we only sample 8192 / k images per batch. Note that this setup is different from the SwAV paper [c] (which is the origin of the idea of > 2 views per image), where they actually increased the total number of crops per batch. The comparison between SimCLR and StableRep is:
>
> | method | pre-train data | 2 view | 4 view | 6 view |
> |--|:--:|:--:|:--:|:--:|
> |SimCLR| Real | 59.5 | 60.0 | 59.3 |
> |StableRep| Syn | **68.7** | **69.6** | **69.6** |
>
> Another ablation is to compare Synthetic SimCLR and StableRep with 2 views. Both are using synthetic images and two views. The difference is that the two views of synthetic SimCLR come from the same image, while the two views of StableRep come from two different images. The comparison shown below clearly demonstrates that most of the improvement comes from using different images of the same caption, rather than just more views.
>
> | method | pre-train data | 2 view |
> |--|:--:|:--:|
> |SimCLR   | Syn | 60.5 |
> |StableRep| Syn | **68.7** |
>
> Please don’t hesitate to let us know if you have more comments or questions. We respectfully hope that you can consider raising the score if our response could address existing concerns.
>
> [a] Unified Perceptual Parsing for Scene Understanding
> [b] MMSegmentation: OpenMMLab Semantic Segmentation Toolbox and Benchmark
> [c] Unsupervised Learning of Visual Features by Contrasting Cluster Assignments

---

> > ### Comment · Reviewer_DyTC · 2023-08-16
> > **Response to rebuttal**
> >
> > Thanks for the authors' detailed response. Most of my concerns have been well addressed. I have one additional question. For the Syn+Real experiments provided in the rebuttal (reviewer HH2w, AnD7), could the authors explain why SimCLR Syn+Real performs worse than SimCLR Real on **few-shot avg**?

---

> > > ### Author Response · Authors · 2023-08-16
> > > **Further response**
> > >
> > > Dear Reviewer DyTC,
> > >
> > > Thank you for your response! This is a good catch, we have checked the results of “Syn+Real” v.s. “Real” SimCLR in the few-shot setting. “Syn+Real” outperforms (or at least on par with) “Real” on all datasets but CIFAR-10/100. On CFAIR-10/100, “Syn+Real” lags significantly behind “Real” (see below), making the overall average accuracy lower. We conjecture this is because of the resolution issue, which we solved for StableRep (check here). So now we are applying the same fix to “Syn+Real” and will update you once the results are available.
> > >
> > > You may have noticed that, differently, “Syn+Real” slightly outperforms “Real” on average accuracy for the few-shot setup. This is because for CIFAR-10/100, “Syn+Real” is on par with “Real”. In terms of why there is such a difference between SimCLR and CLIP behavior, we conjecture it may be because CLIP is a stronger learner so it better captures the low-resolution bias in the real part than SimCLR.
> > >
> > > For your reference, the current few-shot results (without resolution fix) for SimCLR and CLIP on CIFAR-10/100 are as below:
> > >
> > > | method | pre-train data | cifar-10 | cifar-100 |
> > > |--|:--:|:--:|:--:|
> > > |SimCLR| real | 64.0 | 70.4 |
> > > |SimCLR| syn + real | 52.3 | 58.1 |
> > > |CLIP  | real | 77.5 | 82.1 |
> > > |CLIP. | syn + real | 77.5 | 81.6 |

---

> > > > ### Comment · Reviewer_DyTC · 2023-08-17
> > > > **Thanks for the clarification**
> > > >
> > > > Thanks for the clarification. Overall, this paper is interesting and worth sharing with the community. I would like to raise my score to Weak Accept and hope the authors could incorporate the results discussed in the rebuttal in the final version.

---

> > > > > ### Author Response · Authors · 2023-08-19
> > > > > **update on the few-shot results and others**
> > > > >
> > > > > Dear Reviewer DyTC,
> > > > >
> > > > > Thank you for your positive feedback!
> > > > >
> > > > > **1. few-shot accuracy of SimCLR**
> > > > >
> > > > > As promised [here](https://openreview.net/forum?id=xpjsOQtKqx&noteId=HUqqYfDJ89) , we have tried the resolution fix trick [here](https://openreview.net/forum?id=xpjsOQtKqx&noteId=it4GmuE96a) in term of your concern about the few-shot results of SimCLR "Syn + Real" setup. Now we present the full picture of **SimCLR** few-shot results as below:
> > > > >
> > > > > | method | Resolution Fix? | cifar-10 | cifar-100 | aircraft | cars | dtd | flowers | pets | sun397 | caltech101 | food101 | Average |
> > > > > |---|:--:|:-:|:-:|:-:|:-:|:-:|:-:|:-:|:-:|:-:|:-:|:-:|
> > > > > | Real | ❌ | 64.0 | 70.4 | 40.7 | 50.9 | 82.2 | 92.1 | 74.4 | 94.0 | 90.4 | 70.4 | 73.0 |
> > > > > | | ✅ | 72.1 | 76.1 | 40.9 | 51.3 | 81.6 | 91.6 | 74.0 | 93.8 | 90.4 | 69.9 | 74.2 |
> > > > > | Syn | ❌ | 50.0 | 58.9 | 45.2 | 54.2 | 79.8 | 92.0 | 74.6 | 92.9 | 89.1 | 71.0 | 70.8 |
> > > > > | | ✅ | 77.4 | 81.2 | 44.6 | 54.4 | 80.1 | 92.5 | 74.4 | 93.0 | 90.3 | 71.5 | 75.9 |
> > > > > | Syn+Real | ❌ | 52.3 | 58.1 | 44.3 | 54.1 | 83.0 | 92.4 | 76.3 | 94.5 | 91.8 | 72.1 | 71.9 |
> > > > > | | ✅ | 73.5 | 76.4 | 41.6 | 53.4 | 81.6 | 91.9 | 74 | 94.1 | 91.1 | 71 | 74.9 |
> > > > >
> > > > > The findings reveal that implementing the resolution fix augmentation yields a notable improvement in CIFAR-10/CIFAR-100 performance for all three setups. Meanwhile, we notice that this resolution fix will likely result in performance drop on the other 8 datasets for "Real" or "Syn+Real" setup, while in contrast increasing accuracies on "Syn" setup.  As a result, resolution fix significantly increases the average accuracy by 5.1 for "Syn", while 3.0 for "Syn + Real" and only 1.2 for "Real". Consequently, the "Syn" setup attains the highest averaged performance. For future work, one may selectively apply this downsample augmentation (resolution fix strategy) to real images based on their resolution, and potentially could bring "Syn + Real" higher than "Syn".
> > > > >
> > > > > **2. Update on ADE20k segmentation results**
> > > > >
> > > > > Previous results we reported in this thread only adopts a short training schedule (e.g., 20k iterations), and we have switched to the full schedule following MAE (160k iterations). We found our StableRep even outperforms MAE trained on ImageNet under the fully fine-tuning setup.
> > > > >
> > > > > Frozen backbone:
> > > > > |method|pre-train data|mean IoU (%)|pixel acc. (%)|
> > > > > |--|:--:|:--:|:--:|
> > > > > |MAE|ImageNet, real|36.4|77.7|
> > > > > |CLIP|cc12m, real|37.9|78.5|
> > > > > |SimCLR|cc12m, real|39.5|79.5|
> > > > > |SimCLR|cc12m, syn |37.2|78.2|
> > > > > |StableRep|cc12m, syn|**42.3**|**80.4**|
> > > > >
> > > > > Fine-tune:
> > > > > |method|pre-train data|mean IoU (%)|pixel acc. (%)|
> > > > > |--|:--:|:--:|:--:|
> > > > > |Supervised|ImageNet, real|47.4|-|
> > > > > |MoCo v3|ImageNet, real|47.3|-|
> > > > > |MAE|ImageNet, real|48.1|82.9|
> > > > > |CLIP|cc12m, real|45.4|81.7|
> > > > > |SimCLR|cc12m, real|45.9|79.5|
> > > > > |SimCLR|cc12m, syn |44.7|81.9|
> > > > > |StableRep|cc12m, syn|**49.2**|**83.5**|
> > > > >
> > > > > This is interesting and non-trivial result, because:
> > > > >  - previously contrastive learning / joint embedding methods **without masking** strategy were consistently underperforming MAE in dense prediction tasks under the fine-tuning setup. In contrast, our method outperforms MAE
> > > > >  - our method is trained purely with **synthetic** images
> > > > >
> > > > > **3. further summarization**
> > > > >
> > > > > We have presented various results in different threads during the rebuttal and discussion period. We feel some of them might be interesting to you, so we want to share with you a summarization, which can be found [here](https://openreview.net/forum?id=xpjsOQtKqx&noteId=F3K5WcycnW).
> > > > >
> > > > > We would like to know if they can address any of your remained concerns, or enhance your evaluation about our work. We would be very happy if this summarization could lead to a favorable increase of the score, or it just simply makes an enjoyable read. If you have any unsolved/new concerns, feedbacks, or comments stimulated by these additional results, please do not hesitate to let us know.

---

> > > > > ### Author Response · Authors · 2023-08-21
> > > > >
> > > > > Dear Reviewer DyTC,
> > > > >
> > > > > Thank you again for your engagement in this fruitful discussion!
> > > > >
> > > > > As there is only **1 day** left, we just want to check if our latest response has successfully addressed your questions about few-shot evaluation? We respectfully interpret your current rating as that you may still have other concerns or feedbacks. If this is the case, we would appreciate it if you could kindly share them with us, as they may further stimulate improvements either in writing or experiments. We love feedbacks so please don't hesitate!

---

> > > > > > ### Comment · Reviewer_DyTC · 2023-08-21
> > > > > > **Thanks**
> > > > > >
> > > > > > Thanks for the fruitful rebuttal. My concerns about the few-shot evaluation have been well addressed. I would like to finalize my score to 7.

---

> > > > > > > ### Author Response · Authors · 2023-08-21
> > > > > > > **Thank you**
> > > > > > >
> > > > > > > Dear Reviewer DyTC,
> > > > > > >
> > > > > > > Thanks for the positive feedback. We are really glad to learn that our response has addressed your concerns.
> > > > > > >
> > > > > > > Thanks once again for sharing your insightful suggestions and for the dedicated time and effort you spent in reviewing our work!

---

### Official Review · Reviewer_qM14 · 2023-07-04

**Soundness:** 4 excellent
**Presentation:** 4 excellent
**Contribution:** 2 fair
**Rating:** 7
**Confidence:** 5

**Summary:**

The authors presents a novel method for learning visual representations using synthetic data. The authors leverage text-to-image generative models (Stable Diffusion) to synthesize images from textual prompts, which are then used to train a self-supervised visual representation model. The synthetic data generation process is guided by a set of diverse and non-repetitive textual prompts, which helps in creating a wide variety of images. The authors demonstrate that their approach outperforms traditional self-supervised learning methods that rely on real images, especially when the amount of real data is limited. They also propose a multiple positive samples based contrastive learning approach called StableRep to utilize different synthetic images generated from a source text prompt, which outperforms training with simply real or synthetic images on a variety of downstream tasks.

**Strengths:**

1. **Important Problem:** The authors study a very important problem, how to utilize synthetic images for training vision and vision-language models. They also show success at this central problem (with caveats, see weaknesses). They are one of the first studies showing successful use of synthetic data from generative models on standard computer vision benchmarks.

2. **Novel approach:** The multi-positive StableRep approach proposed by the authors for multi-caption invariance learning in self-supervised models is a novel idea and seems to work very well in practice leading to significant improvements over simply using synthetic images for training.

3. **Ablations:** The authors perform extensive ablations on the role of guidance scale, and also include a study on how additional language supervision could be used to increase caption efficiency while training with multiple positive synthetic images. The results indicate that proposed approaches (StableRep and StableRep+) indeed are more efficient versus training on real images.

**Weaknesses:**

1. **Limited motivation behind why synthetic images:** It is unclear to me as a reader why Synthetic Images are being used in Section 2.2 as training data for pre-training. It seems that there is no real benefit from using them on state-of-the-art contrastive models like BYOL and MoCo-V3 in terms of linear accuracy. Plus, the key issue for training is the lack of real world labelled data as mentioned by the authors' central question "how can we collect such large amounts of varied data to train AI models?" (L21). There is plenty of real-world unlabelled data that is available and can be used to train self-supervised models. The real benefit that synthetic images present is unlimited labelled data. However no fine-tuning comparisons or supervised learning comparisons are provided except for fine-tuning MAE, which could very well be explained by the stochasticity in model training (82.9% vs 82.6%). Section 2.2 does not do much to justify the usage of synthetic images for training in a manner that addresses the authors' central question, and follow up sections justify it for efficiency but not performance.

2. **Limits of generative model:** While the authors include a limitation section, and mention briefly the issues that affect generative models like Stable Diffusion, there needs to be more rigorous evaluation of these issues. Conceptually, these methods suffer from issues like limited compositional understanding. There have been countless studies on the social biases reflected in these models. Very lately (after the NeurIPS deadline), there was an important work showing that models trained on data from generative models can suffer from model collapse, where tails of the original data distribution disappear from the subsequent trained model [1]. All these issues are quite major and should be part of this study, since it is one of the first to utilize synthetic data and show some improvements in results and could potentially effect future research in this area. In particular, the authors should include results on compositional benchmarks (like ARO, CREPE, Winoground, SugarCREPE etc) for their CLIP models trained with StableRep, discuss the fairness of the self-supervised models such as worst-class accuracy and geographic bias etc.

### References

1. Shumailov I, Shumaylov Z, Zhao Y, Gal Y, Papernot N, Anderson R. The Curse of Recursion: Training on Generated Data Makes Models Forget. arXiv preprint arXiv:2305.17493. 2023 May 27.

**Questions:**

My main concern is with the motivation behind using synthetic images as well as the limited commentary and no evaluation of the downstream effects of using data from generative models for training. I would be willing to improve my rating if the authors address these issues (as discussed in weaknesses).

**Limitations:**

The authors have addressed the limitations of their work in Section 7, but it could use further commentary.

---

> ### Author Rebuttal · Authors · 2023-08-10
>
> Thank you for recognizing our work as novel and acknowledging that we study a very important problem.
>
> > Limited motivation behind why synthetic images: It is unclear to me as a reader why Synthetic Images are being used in Section 2.2 as training data for pre-training. It seems that there is no real benefit from using them on state-of-the-art contrastive models like BYOL and MoCo-V3 in terms of linear accuracy
>
> The whole point of Section 2.2 is to analyze and study how good synthetic images are for self-supervised training, thus is **not** about chasing numbers. It's actually quite surprising to find that using synthetic images are on par at (or often better than) real images. We have improved the training for MoCo v3, and now the results are even closer between real and synthetic images (see below). This is an surprising finding, not to mention that synthetic images are much better than real images for MAE for linear probing.
>
> |  | data | cc3m | cc12m|
> |--|:--:|:--:|--:|
> | MoCo v3 | real | 64.8 | 66.6 |
> |         | syn  | 64.3 | 66.4 |
> | BYOL    | real | 64.0 | 65.9 |
> |         | syn  | 64.2 | 65.6 |
>
> Learning from synthetic data is quite and interesting topic, and there actually have been a whole line of work exploring synthetic images for training (such as [3],[24],[58],[31]). One advantage of using synthetic data is that we have multiple ways of freedom to control the way we generate the data. Sec 2.2 shows one freedom the guidance scale, while Sec 2.3 shows another freedom which is leveraging the stochasticity to generate multiple images per caption. We hope our work can inspire future exploration in this direction.
>
> > no supervised learning comparisons
>
> We have include supervised learning comparisons in the question 1 of the "global rebuttal" shared to all reviewers. Feel free to let us know if you mean something else.
>
> > no fine-tuning comparisons
>
> We add additional ImageNet fine-tuning results for SimCLR:
> |  | Pre-train data | ImageNet fine-tuning |
> |--|:--:|:--:|
> | SimCLR | Real | 81.8|
> |        | Syn | **82.2**|
>
> We tried Stable Diffusion 2.1 for synthetic images, and SimCLR and MAE with SD 2.0 achieved 82.1 and 83.0 ImageNet fine-tuning, both higher than pre-train on the real images (81.8 for SimCLR and 82.6 for MAE)
>
> > which could very well be explained by the stochasticity in model training (82.9% vs 82.6%)
>
> Not really, we fine-tuned multiple times and the averaged results remain the same. The performance gap in fine-tuning is usually small, please refer to Table 3 in the MAE paper to see that MAE also only beat MoCo v3 and BEiT by 0.4%.
>
> > Section 2.2 does not do much to justify the usage of synthetic images for training in a manner that addresses the authors' central question, and follow up sections justify it for efficiency but not performance.
>
> We believe we have justified the performance in our paper (acknowledged by other reviewers too). Also see question 1 in "global rebuttal".
>
> > All these issues are quite major and should be part of this study, since it is one of the first to utilize synthetic data and show some improvements in results and could potentially effect future research in this area. In particular, the authors should include results on compositional benchmarks (like ARO, CREPE, Winoground, SugarCREPE etc) for their CLIP models trained with StableRep, discuss the fairness of the self-supervised models such as worst-class accuracy and geographic bias etc.
>
> This is a great suggestion! We tried ARO benchmark for compositionality test, and FairFace for evaluating representation fairness. We generally found that training on Synthetic data is able to improve the performance on both compositionality and fairness.
>
> For ARO (compositionality):
> |  |pre-train data| Relation accuracy |
> |--|:--:|:--:|
> | CLIP | Real | 46.4 |
> |      | Syn  | **50.0** |
> | StableRep+ | Syn | 47.3 |
>
> For FairFace (fairness):
> |  |pre-train data| mean accuracy | best-class accuracy | worst-class accuracy |
> |--|:--:|:--:|:--:|:--:|
> | CLIP | Real | 28.2 | 60.2 | 0.3 |
> |      | Syn  | 30.4 | 64.0 | 3.1 |
> | StableRep+ | Syn | **37.2** | **74.9** | **10.0** |
>
> CLIP w/ real data only achieved 0.3% accuracy with "southeast asian male" class, and CLIP w/ synthetic data improves this class to 3.1%, while our StableRep+ improves this class to 27.2%. The worst class for StableRep+ is "middleastern male", which CLIP w/ real and w/ synthetic struggle too (they achieved 6.9% and 6.2%). This suggests a geographic bias and StableRep+ generally improves the fairness.
>
> Please don’t hesitate to let us know for any additional comments on the paper or the ARO or FairFace experiments.

---

> > ### Comment · Reviewer_qM14 · 2023-08-14
> > **Response to rebuttal**
> >
> > Thank you for taking the time to write a thorough rebuttal, and also running the additional experiments based on my suggestions. The new results indeed solidify the utility of StableRep, and I am happy to say my concerns have largely been addressed. I have increased my rating from a Weak Accept to an Accept.

---

> > > ### Author Response · Authors · 2023-08-16
> > > **Thanks**
> > >
> > > Dear Reviewer qM14,
> > >
> > > Thank you for reading our response! We are happy to see it largely addressed your concerns. For any further discussions/experiments/clarifications that you think are able to address your remaining concerns, please feel free to let us know!

---

> > > ### Author Response · Authors · 2023-08-19
> > > **Update of additional (interesting) results**
> > >
> > > Dear Reviewer qM14,
> > >
> > > Thank you again for your engagement in the discussion!
> > >
> > > We have presented various results in the rebuttal to other reviewers in addition to you, and obtained some new results after the rebuttal deadline. We believe some of them might be interesting to you, so we want to share with you a summarization, which can be found [here](https://openreview.net/forum?id=xpjsOQtKqx&noteId=F3K5WcycnW).
> > >
> > > We would be very happy if they can address any of your remained concerns, or enhance your evaluation about our work, or simply just let you feel our work is more interesting. Meanwhile, should you have any unsolved/new concerns or feedbacks stimulated by these additional results, please don't hesitate to let us know.

---

### Official Review · Reviewer_AnD7 · 2023-07-08

**Soundness:** 3 good
**Presentation:** 4 excellent
**Contribution:** 3 good
**Rating:** 7
**Confidence:** 4

**Summary:**

This paper investigates how synthetic data generated with the text-to-image diffusion model Stable Diffusion can be leveraged for representation learning.
To this end, the paper analyzes established representation learning approaches such as SimCLR and CLIP, but trained on the synthetically generated data. Further, it introduces a novel method,
dubbed "StableRep", specifically designed for representation learning from a generative model, to generate multiple "positive" image instances for downstream representation learning.
Experiments demonstrate the potential of synthetic data for representation learning and mostly perform on-par with or better than when trained on real data only.

**Strengths:**

This is a well-written paper, which approaches a timely research question: To what extent can large-scale, generative models which have been trained on internet-scale data be leveraged for other tasks, by augmenting or replacing real data. The experiments are well-designed and hint at the potential of using synthetic data. StableRep is a nice method specificially designed for generative models that can generate multiple positive examples for a given prompt. The experiments are encouraging for future research in this area.

**Weaknesses:**

While experiments demonstrate potential, there remain a few unaddressed points: How does the choice of generative model influence the results for representation learning? In particular, Stable Diffusion is conditioned on CLIP text features, which themselves are pretrained through CLIP's contrastive objective. There exist, however, other generative text-to-image variants that use non-contrastive features for conditioning, such as T5-representations (e.g., Imagen [https://github.com/deep-floyd/IF], IF [https://github.com/deep-floyd/IF]).
Further, the paper does not discuss how the size of the pretraining dataset for both CLIP (for conditioning SD) and the training data for SD change performance of StableRep (the pretraining data is much larger than CC3M/12M). In addition, the possibility of data poisoning w.r.t to the results presented in Tab. 2 and Tab. 3 is not discussed (for example, SD's training data might have contained examples from the linear probing datasets). The performance gap between CIFAR-10/100 and the other datasets remains unexplained.
l. 1014-107 need more detail on hyperparameter settings like number of steps, learning rate used, convergence of the model.

For further comments, please refer to the "Questions" section.

**Questions:**

- Can we mix synthetic and real data? How does this affect the outcome of training approaches like CLIP and SimCLR or StableRep?
- Other sampling hyperparameters than guidance scale w are not discussed. How does the choice of solver change the results? Same question for the number of denoising steps / function evaluations...
- small note: In Eq. 1, t could be misinterpreted as timestep variable, as often used in the context of diffusion models.
- Sec. 4, l. 229-233: Do the instabilities for larger models only occur for synthetic data?
- Sec. 5, l. 240-245: This section speculates about the inferior results when training CLIP on synthetic data only (vs real data only). Why not plot the imagenet zero-shot accuracy vs clip-image similarity for different guidance scales? Using a better model than SD 1.5 might also help clarify this point.

**Limitations:**

yes.

---

> ### Author Rebuttal · Authors · 2023-08-10
>
> Thank you for acknowledging our method is nice and that our experiments are encouraging, and we appreciate the constructive feedback. We address your concerns or questions one by one as below.
>
> > How does the choice of generative model influence the results for representation learning?
>
> This is a good point, and we provide the results of using DeepFloyd-IF instead. Given that DeepFloyd-IF is slow and rebuttal time is limited, we: (1) only used its stage 1 & 2 models; (2) never tuned its hyper-perameter. We train StableRep with 2 views and and present the results as below:
>
> | method | ImageNet LP |
> |--|:--:|
> | StableRep w/ SD | 65.8 |
> | StableRep w/ IF | 65.5 |
>
> This suggests StableRep also works with models that are *not* conditioned on CLIP features.
>
> > Further, the paper does not discuss how the size of the pretraining dataset for both CLIP (for conditioning SD) and the training data for SD change performance of StableRep.  In addition, the possibility of data poisoning w.r.t to the results presented in Tab. 2 and Tab. 3 is not discussed.
>
> Unfortunately, we cannot train CLIP and Stable Diffusion (SD) with various different data sizes to ablate this. In terms of poisoning we perform an surrogate sanity check: we compute how close the downstream datasets are to the StableRep's training set. Specifically, for an image $I$ in a downstream dataset, we find its closest image in StableRep's training set by using consine similarity of some pre-trained SimCLR feature extractor $f$ (Recall that SimCLR is optimized to maximize cosine similarity of the same image). We define this highest similarity ($s$) as this image's similarity with StableRep's training set $D$. Concretely, $s_i = \max_{X \in D} cos(f(I_i), f(X))$. We evaluate four datasets, and provide the statistics (avg, min, and max) of this highest similarity metric of each image.
>
> | Train Set | stats | aircraft|cars|flowers|food101|
> |--|:--:|:--:|:--:|:--:|:--:|
> | StableRep Train | avg | 0.891 | 0.836 | 0.836 | 0.791 |
> || min | 0.652 | 0.258 | 0.523 | 0.443 |
> || max | 0.985 | 0.980 | 0.967 | 0.958 |
> | ImageNet  Train | avg | 0.901 | 0.870 | 0.823 | 0.807 |
> || min | 0.700 | 0.629 | 0.586 | 0.560 |
> || max | 0.980 | 0.998 | 0.988 | 0.977 |
>
> The results show downstream datasets are closer to ImageNet (which has been widely used for pre-training and then evaluating on these downstream datasets) than to StableRep's train set, therefore we believe the training set of StableRep is not poisoned.
>
> Please refer to our response to reviewer FSjQ for more relevant discussions.
>
> > The performance gap between CIFAR-10/100 and the other datasets remains unexplained
>
> We have identified this is because CIFAR-10/100 have very low resolution images. More detailed results and explanations are provided in the question 2 of the "global rebuttal" (please find it around the top of this page).
>
> > Can we mix synthetic and real data?
>
> Yes.
>
> For SimCLR:
> | method | pre-train data | cifar-10 | cfiar-100 | ImageNet LP | Downstream avg | few-shot avg|
> |--|:--:|:--:|:--:|:--:|:--:|:--:|
> | SimCLR | Real | 88.3 | 70.3 | 61.5 | 72.3 | **73.0** |
> | SimCLR | Syn  | 84.8 | 65.2 | 63.7 | 72.7 | 70.8 |
> | SimCLR | Syn+Real | **88.7** | **71.9** | **64.1** | **74.9** | 71.9 |
>
> For CLIP:
> | method | pre-train data | cifar-10 | cfiar-100 | ImageNet LP | Downstream avg | few-shot avg|
> |--|:--:|:--:|:--:|:--:|:--:|:--:|
> | CLIP | Real | **94.0** | 79.0 | 70.3 | 81.2 | 86.7 |
> | CLIP | Syn | 87.3 | 69.5 | 67.8 | 79.1 | 83.7 |
> | CLIP | Syn+Real | 93.9 | **80.2** | **73.3** | **83.0** | **87.3** |
>
> For StableRep (StableRep only trained with 2 views and 15 epochs):
> | method | pre-train data | cifar-10 | cfiar-100 | ImageNet LP | Downstream avg | few-shot avg|
> |--|:--:|:--:|:--:|:--:|:--:|:--:|
> | StableRep (2 views) | Syn | 88.5 | 71.5 | 67.9| 78.0 | 81.4 |
> | StableRep (2 views) | Syn+Real | **92.2** | **76.1** | **69.1** | **79.3** | **84.2** |
>
> Combining synthetic and real datasets uniformly improve all methods
>
>
> > -Other sampling hyperparameters than guidance scale w
>
> This is a good suggestion, and the comparison are as below:
>
> | method | solver | step | ImageNet LP|
> |--|:--:|:--:|:--:|
> | baseline | DDIM  | 50  | 65.8 |
> || DDIM | 25  | 64.9 |
> || DDIM | 100 | 65.8 |
> || DPM | 50  | 66.0 |
>
> Halving the sample steps brings about 0.9% drop, while doubling the steps has no effect. Switching from the ODE solver DDIM to SDE solver DPM slightly improves over the baseline by 0.2%.
>
> > Do the instabilities for larger models only occur for synthetic data?
>
> No. Large vision transformers are notoriously unstable during training in general. This paper [a] has more discussion about it.
>
> > This section speculates about the inferior results when training CLIP on synthetic data only (vs real data only). Why not plot the imagenet zero-shot accuracy vs clip-image similarity for different guidance scales?
>
> We worry that plot w/ clip score can be misleading. The performance of CLIP trained with synthetic images drops when guidance scale increases, and meanwhile clip score increases with guidance scale. Putting togther, it becomes that the higher the clip score, the lower the CLIP performance. This is misleading and only part of the story. We know that guidance scale trades off between data diversity and image-text alignment. While image-text alignment increases with guidance scale, data diversity level reduces a lot. As a result, the disadvantage brought by diversity dropping outweighs the advantage brought by better image-text alignment.
>
> We are more than happy to discuss more if you have further questions. We hope our response will encourage the reviewer to consider increase the rating if they have addressed your concern.
>
> [a] Scaling Vision Transformers to 22 Billion Parameters

---

> ### Author Response · Authors · 2023-08-19
> **Any further questions/concerns are more than welcome**
>
> Dear Reviewer AnD7,
>
> We would like to thank the reviewer again for your time and effort. Except for rebuttal in this thread, we have made a summarization of new (and maybe interesting) results, check [here](https://openreview.net/forum?id=xpjsOQtKqx&noteId=F3K5WcycnW). We believe this evidence, together with our initial rebuttal should be able to address most of your concerns (if not all).
>
> Given there is only  **< 3 days left**  in the discussion period, we want to let you know that we would love to take any additional questions/comments. Meanwhile, if the concerns of the reviewer are clarified and the reviewer is convinced of the novelty and completeness of our work, we'd be grateful if the reviewer could update your review to reflect that. Once again, many thanks for your time and dedication to the review process, we are extremely thankful!

---

> > ### Comment · Reviewer_AnD7 · 2023-08-19
> > **Great Rebuttal**
> >
> > Dear Authors,
> >
> > Thank you for the work you put into this response and the general response. I think this is a strong rebuttal that answers most of my questions. I would like to see this work at NeurIPS 2023 and raise my score to 7.

---

> > > ### Author Response · Authors · 2023-08-21
> > > **Thank you**
> > >
> > > Dear Reviewer AnD7,
> > >
> > > We appreciate your positive feedback. We are truly delighted to see that our response has addressed your questions.
> > >
> > > Thank you once again for dedicating your time and effort to reviewing our work and providing us with insightful suggestions!

---

### Official Review · Reviewer_HH2w · 2023-07-09

**Soundness:** 4 excellent
**Presentation:** 4 excellent
**Contribution:** 3 good
**Rating:** 5
**Confidence:** 4

**Summary:**

This paper studies pre-training by using generated image from diffusion models. It presents StableRep that generate different images with the same caption by using stable diffusion models. The model is hence pre-trained by using the generated samples and contrastive loss. Extensive experiments demonstrate the effectiveness of the proposed method by using synthetic images only.

**Strengths:**

1. This paper studies the pre-training by using synthetic data only. Experimental results show that pre-training by using synthetic data only outperforms the pre-training over real images.
2. This paper is generally clear and well-written.

**Weaknesses:**

The pre-training over generated images by using diffusion models have been studied in [24]. In [24], it seems direct pre-training on synthetic images could helps to improve the classification results without the proposed pre-training pipeline. As the comparison between the proposed method and the reported results in [24] is missing, it is difficult the determine the effectiveness of the proposed pre-training pipeline.

**Questions:**

1. For general vision-language pre-training methods, image labels are usually very difficult to obtain. This paper studies the use of synthetic data, where the labels of synthetic images can be obtained during image synthesis. As such, we can simply pre-train the model by generating images together with the labels in the task-specific label spaces, which could improve the model performances [24]. As such, it is not clear why the contrastive pre-training is necessary or better when using synthetic data for pre-training. More discussion and experiments are expected.
2. According to the experimental results, pre-training on synthetic data helps to improve the performances on most of the datasets while may introduce degradation some of the datasets (e.g. CIFAR-10). How about combining the synthetic and real datasets together for pre-training?

**Limitations:**

Yes

---

> ### Author Rebuttal · Authors · 2023-08-10
>
> Thank you for the review. In the following we address your concern in detail.
>
> > the comparison between the proposed method and the reported results in [24] is missing
>
>
> This is a reasonable comparison, which we have included in the question 1 of "global rebuttal" shared to all reviewers (please find it around the top of the page).
>
> Our StableRep clearly outperforms [24] (and additionally [a]), and even outperforms supervised training using Real ImageNet on both downstream transfer and few-shot evaluation.
>
> > As such, we can simply pre-train the model by generating images together with the labels in the task-specific label spaces, which could improve the model performances [24]. As such, it is not clear why the contrastive pre-training is necessary or better when using synthetic data for pre-training. More discussion and experiments are expected.
>
> Firstly, as shown in aforementioned experiments, our StableRep has higher accuracy than [24], despite that we do not specifically use ImageNet labels.
>
> Secondly, using target label space for synthesis means we have to know the downstream task beforehand. This means we have to synthesize a different dataset and train a different model for each task, which clearly limits the applicability of this approach. In contrast, we are aiming at learning a general purpose representation which hopefully can be easily transferred to a broader range of tasks. The aforementioned comparison between [24] and our StableRep also suggests that our representation has stronger transferability.
>
> > According to the experimental results, pre-training on synthetic data helps to improve the performances on most of the datasets while may introduce degradation some of the datasets (e.g. CIFAR-10)
>
> We have identified this issue. This is due to the image resolution: cifar-10/100 has very low resolution image (32x32 pixels), while other evaluation datasets have much higher resolutions. To test this hypothesis, we also downsampled the other 9 evaluation datasets to 32x32 pixels before we evaluate. The results and explanations are presented in question 2 of the "global rebuttal" (please find it around the top of the page). A fix may be to randomly downsample the synthetic image to very low resolution during StableRep training, which we leave as future work given the limited rebuttal time.
>
> > How about combining the synthetic and real datasets together for pre-training?
>
> This is a good suggestion, and the results of combining synthetic and real datasets are as below.
>
> For SimCLR:
> | method | pre-train data | cifar-10 | cfiar-100 | ImageNet LP | Downstream avg | few-shot avg|
> |--|:--:|:--:|:--:|:--:|:--:|:--:|
> | SimCLR | Real | 88.3 | 70.3 | 61.5 | 72.3 | **73.0** |
> | SimCLR | Syn  | 84.8 | 65.2 | 63.7 | 72.7 | 70.8 |
> | SimCLR | Syn+Real | **88.7** | **71.9** | **64.1** | **74.9** | 71.9 |
>
> For CLIP:
> | method | pre-train data | cifar-10 | cfiar-100 | ImageNet LP | Downstream avg | few-shot avg|
> |--|:--:|:--:|:--:|:--:|:--:|:--:|
> | CLIP | Real | **94.0** | 79.0 | 70.3 | 81.2 | 86.7 |
> | CLIP | Syn | 87.3 | 69.5 | 67.8 | 79.1 | 83.7 |
> | CLIP | Syn+Real | 93.9 | **80.2** | **73.3** | **83.0** | **87.3** |
>
> For StableRep (StableRep items here only use two views per caption, and are trained with only 15 epochs):
> | method | pre-train data | cifar-10 | cfiar-100 | ImageNet LP | Downstream avg | few-shot avg|
> |--|:--:|:--:|:--:|:--:|:--:|:--:|
> | StableRep (2 views) | Syn | 88.5 | 71.5 | 67.9| 78.0 | 81.4 |
> | StableRep (2 views) | Syn+Real | **92.2** | **76.1** | **69.1** | **79.3** | **84.2** |
>
> With these results, we conclude:
> - the cifar-10/100 resolution issue causes both CLIP and SimCLR to drop if only trained on synthetic data, but combining both synthetic and real datasets significantly alleviate this issue (and often surpass real-only).
> - combining synthetic and real datasets uniformly improve all methods
>
> We hope that these discussion and results help address your concern and lead to a favorable increase of the score. Please don’t hesitate to let us know for any additional comments or suggestions.
>
>
> [a] Fake it till you make it: Learning transferable representations from synthetic ImageNet

---

> ### Author Response · Authors · 2023-08-19
> **Any further questions/concerns are more than welcome**
>
> Dear Reviewer HH2w,
>
> We would like to thank the reviewer again for your time and effort. Except for rebuttal in this thread, we have made a summarization of new (and maybe interesting) results, check [here](https://openreview.net/forum?id=xpjsOQtKqx&noteId=F3K5WcycnW). We believe this evidence, together with our initial rebuttal should be able to address all your concerns.
>
> Given there is only  **< 3 days left**  in the discussion period, we want to let you know that we would love to take any additional questions/comments. Meanwhile, if the concerns of the reviewer are clarified and the reviewer is convinced of the novelty and completeness of our work, we'd be grateful if the reviewer could update your review to reflect that. Once again, many thanks for your time and dedication to the review process, we are extremely grateful.

---

### Author Rebuttal · Authors · 2023-08-10

We thank all reviewers for their insightful comments and feedback! We are glad to see the reviewers found:

 1. the paper is well-written and easy to follow (by reviewer HH2w, AnD7, DyTC)
 2. this paper approaches an interesting timely research question / very important problem (by AnD7, DyTC, FSjQ, qM14).
 3. the approach is novel (by qM14, FSjQ) and nice (by AnD7).
 4. this paper shows well-designed experiments / extensive ablation study / comprehensive evaluation / encouraging and promising results (by all reviewers).


We provide answers to shared questions/concerns here (and respond to remaining individual questions to each reviewer separately):


***1. directly use labels (e.g. from ImageNet) to synthesize image?***


We acknowledge that the challenging part of synthesizing with labels is how to design the text prompts. We include four variants here:

- **Language Enhancement (LE)** proposed by [24]: it generates 200 prompts for each ImageNet class.
-  **c, hc, in place** proposed by [a]: it composes ImageNet class name *c* with the hypernyms of the class *hc* and Places class name *place*.
- **c, dc, in place** proposed by [a]: it composes ImageNet class name *c* with the definition of the class *dc* and Places class name *place*.
- **a photo of [category]** proposed by us (also suggested by Reviewer DyTC).

For each of above items, we perform both supervised cross-entropy (xent) training and SimCLR training. We compare them with our StableRep, as well as training on the Real ImageNet dataset. All methods are trained for roughly 300 ImageNet-equivalent epochs for a fair comparison. We report: (1) linear probing accuracy on ImageNet (ImageNet LP); (2) the average accuracy of the linear transfer on 12 downstream datasets; (3) the average accuracy of few-shot evaluation on 11 datasets. Results are as follows:

|  | pre-train data | method | ImageNet LP | Downstream avg | few-shot avg|
|--|:--:|:--:|:--:|:--:|:--:|
| Supervised | Real ImageNet | xent | - | 78.6 | 85.5 |
| Unsupervised | Real ImageNet | SimCLR | **74.3** | 74.3 | 77.8 |
| LE [24] | Synthetic | xent | 64.7 | 71.5 | 77.5 |
|    | Synthetic | SimCLR | 66.8 |74.0 | 71.9 |
| c, hc, in place | Synthetic | xent | 67.1 |73.0 | 77.2 |
|  | Synthetic | SimCLR | 66.1 | 73.2 | 71.1 |
| c, dc, in place | Synthetic | xent | 68.6 | 73.2 | 77.2 |
|  | Synthetic | SimCLR | 65.8 | 73.2 | 69.2 |
| a photo of c | Synthetic | xent | 67.9 | 72.6 | 76.5 |
|  | Synthetic | simclr | 66.6 | 74.1 | 71.3 |
| StableRep | Synthetic | StableRep |72.8 | **82.2** | **85.9** |


We observe that:
- Our StableRep clearly outperforms all models trained on synthetic imagenet (even on ImageNet linear probing despite synthetic ImageNet explicitly leverages ImageNet labels), and even outperforms supervised/unsupervised training using Real ImageNet on both downstream transfer and few-shot evaluation.
- StableRep is only inferior to SimCLR trained with ImageNet on the ImageNet linear probing benchmark, mostly because StableRep here hasn't converged yet. In fact, our convergent StableRep with longer training can reach 75.7 (using cc12m captions) or 76.7 (using RedCaps captions). In contrast, SimCLR with real ImageNet does not improve much with longer training, indicating the item here is already convergent.


***2. StableRep (or training with synthetic data) underperforms training with real data on cifar-10/100 evalution, while outperforms on the other evaluation datasets ?***


We have identified this issue. This is due to the image resolution: cifar-10/100 has very low resolution image (32x32 pixels), while other evaluation datasets have much higher resolutions. To test this hypothesis, we also downsampled the other 9 evaluation datasets to 32x32 pixels before we evaluate. The results for our StableRep and CLIP with real data are as below:

| method | downsample? | aircraft | cars| dtd| flowers| pets| sun397| caltech101 | food101 | voc2007| Average |
|--|:--:|:--:|:--:|:--:|:--:|:--:|:--:|:--:|:--:|:--:|:--:|
| real CLIP |  | 53.2| 75.8 | 75.7 | 96.0 | 86.7 | 72.5 | 92.7 | 81.6 | 86.1 | **80.0** |
| real CLIP | ✅ | 10.2| 35.7 | 48.7 | 71.6 | 70.4 | 58.5 | 77.9 | 66.8 | 83.5 | **58.1** |
| Stablerep |  | 57.6| 80.3 | 79.0 | 96.7 | 87.1 | 73.2 | 94.0 | 83.5 | 87.2 | **82.1** |
| Stablerep | ✅ | 5.2 | 24.3 | 53.3 | 53.2 | 65.4 | 46.2 | 70.7 | 52.0 | 83.3 | **50.4** |


While StableRep outperforms Real CLIP on all of these 9 tasks before downsampling, it underperforms Real CLIP on 8 out 9 tasks (sometimes significantly) after the downsampling. This can be explained by the training data. While Stable Diffusion only synthesizes high resolution images for StableRep training, the real image set often contain many low-resolution or blurred images which in turn help real CLIP's transfer on data of very low resolutions, e.g., 32x32 for cifar-10/100. A fix may be to randomly downsample the synthetic image to very low resolution during StableRep training, which we leave as future work given the limited rebuttal time.

Please don’t hesitate to let us know for any additional feedback. Thanks!


[a] Fake it till you make it: Learning transferable representations from synthetic ImageNet

---

> ### Author Response · Authors · 2023-08-14
> **Fix the cifar-10/100 performance caused by resolution gap**
>
> Dear Reviewers and AC,
>
> We have fixed the performance gap on CIFAR-10/100 caused by resolution issue, and now present the updated linear-probing / few-shot results (corresponding to Table 2 & 3 results in paper) here:
>
> - Linear probing:
>
> | method | cifar-10 | cifar-100 | aircraft | cars| dtd| flowers| pets| sun397| caltech101 | food101 | voc2007| Average |
> |--|:--:|:--:|:--:|:--:|:--:|:--:|:--:|:--:|:--:|:--:|:--:|:--:|
> | real SimCLR | 88.3 | 70.3 | 47.1 | 45.5 | 76.2 | 92.5 | 70.1 | 65.4 | 83.8 | 75.0 | 81.2 | 72.3 |
> | real CLIP | 94.0 | 79.0 | 53.2| 75.8 | 75.7 | 96.0 | **86.7** | 72.5 | 92.7 | 81.6 | 86.1 | 81.2 |
> | Stablerep | **96.4** | **83.8** | **56.8** | **80.5** | **79.4** | **97.2** | **86.8** | **72.9** | **94.3** | **83.3** | **87.1** | **83.5**
>
> - few-shot evaluation:
>
> | method | cifar-10 | cifar-100 | aircraft | cars| dtd| flowers| pets| sun397| caltech101 | food101 | Average |
> |--|:--:|:--:|:--:|:--:|:--:|:--:|:--:|:--:|:--:|:--:|:--:|
> | real SimCLR | 64.0 | 70.4 | 40.7 | 50.9 | 82.2 | 92.1 | 74.4 | 94.0 | 90.4 | 70.4 | 73.0
> | real CLIP | 77.5 | 82.1 | **62.0** | 90.9 | 83.3 | 97.6 | 91.1 | **97.2** | 98.2 | **87.0** | 86.7 |
> | Stablerep | **92.7** | **92.0** | 60.4 | **91.8** |**86.4** | **98.2** | **91.9** | **97.3** | **98.8** | **87.1** | **89.7**
>
>
> What we tried is, during StableRep training, we randomly downsampled the image to a low resolution (64 or 128) and upsampled it to 256x256 (mimicking if we run into low resolution images). We prepended this augmentation to normal augmentation with a probability of $p=0.05$.
>
> After this augmentation, the performance on CIFAR-10/100 gets improved a lot, while the accuracy on the other datasets or ImageNet remains similar as before.

---

### Author Response · Authors · 2023-08-19
**Summary of additional (interesting) results**

Dear reviewers,

We thank you again for providing lots of insightful and constructive feedbacks! Because of that, we are able to update many additional (and perhaps interesting) results during the rebuttal time. We present the **key** summaries here, eliminating the need for each reviewer to constantly reference other responses.

**1. We  identified the performance issue on cifar-10/100, and proposed a simple fix. Now StableRep consistently outperforms others**

Please refer to [here](https://openreview.net/forum?id=xpjsOQtKqx&noteId=it4GmuE96a).

**2. Our method significantly improves the fairness of the representations.**

Thanks to reviewer qM14's suggestion, we evaluate our method on the FairFace benchmark.

The comparison is:
||pre-train data|mean accuracy|best-class accuracy|worst-class accuracy|
|--|:--:|:--:|:--:|:--:|
|CLIP|Real|28.2|60.2|0.3|
||Syn|30.4|64.0|3.1|
|Ours |Syn|**37.2**|**74.9**|**10.0** |

CLIP w/ real data only achieved 0.3% accuracy with "southeast asian male" class, and CLIP w/ synthetic data improves this class to 3.1%, while our method improves this class to 27.2%. The worst class for ours is "middleastern male", which CLIP w/ real and w/ synthetic struggle too (they achieved 6.9% and 6.2%). This suggests a geographic bias and our method generally improves the fairness.

**3. Our method also improves the understanding of compositionality.**

We evaluate on the ARO (Attribute, Relation, and Order) benchmark:
||pre-train data| Relation accuracy |
|--|:--:|:--:|
| CLIP| Real|46.4|
|| Syn|**50.0**|
|Ours| Syn |47.3|

While our method slightly improves over CLIP w/ Real, CLIP w/ Syn surprisingly outperforms CLIP w/ Real by 3.6, highlighting the potential of using synthetic data.

**4. Our method performs well on dense prediction task, and even outperforms the official MAE checkpoint**

Following suggestion from reviewer DyTC, we evaluated representations on ADE20k segmentation dataset.  We follow the original MAE paper to use a UperNet, and used the mmsegmentation library following their optimized parameters. We trained for the full-schedule (160k steps), and considered two setup: (1) frozen the pre-trained part; and (2) fine-tune the whole network. We present the results as below:

Frozen:
|method|pre-train data|mean IoU (%)|pixel acc. (%)|
|--|:--:|:--:|:--:|
|MAE|ImageNet, real|36.4|77.7|
|CLIP|cc12m, real|37.9|78.5|
|SimCLR|cc12m, real|39.5|79.5|
|SimCLR|cc12m, syn |37.2|78.2|
|StableRep|cc12m, syn|**42.3**|**80.4**|

Fine-tune:
|method|pre-train data|mean IoU (%)|pixel acc. (%)|
|--|:--:|:--:|:--:|
|Supervised|ImageNet, real|47.4|-|
|MoCo v3|ImageNet, real|47.3|-|
|MAE|ImageNet, real|48.1|82.9|
|CLIP|cc12m, real|45.4|81.7|
|SimCLR|cc12m, real|45.9|79.5|
|SimCLR|cc12m, syn |44.7|81.9|
|StableRep|cc12m, syn|**49.2**|**83.5**|

We observe that:
(1) StableRep consistently outperforms all other methods trained with real/synthetic data from cc12m/ImageNet dataset.
(2) In the fully fine-tuning setup, StableRep even outperforms MAE trained with real ImageNet dataset. This is interesting and non-trivial results, because:
 - previously contrastive learning methods **without masking** strategy were consistently underperforming MAE in dense prediction tasks under the fine-tuning setup. In contrast, our method outperforms MAE
 - our method is trained purely with **synthetic** images

**5. StableRep generalizes well to ImageNet**

We fine-tune various pre-trained models on ImageNet, hinted by qM14. The results are presented as follow:

ImageNet fine-tuning results:
|method|pre-train data|Top-1 Acc. (%)|
|--|:--:|:--:|
|DINO|ImageNet, real|82.8|
|MoCo v3|ImageNet, real|83.2|
|StableRep|cc12m, syn|**83.5**|

We note that StableRep performs better than other contrastive/Joint-embedding approaches (DINO, MoCo v3) pre-trained directly on ImageNet, despite the domain gap between cc12m and ImageNet. This shows that our StableRep possesses strong generalization ability.

**6. Synthetic data can be unified with real data to further improve the performance**

Results w/o resolution fix yet:
|method|pre-train data|ImageNet LP|Downstream avg|few-shot avg|
|-----------|:--------------:|:------:|:------:|:------------:|
|SimCLR|Real|61.5|72.3|**73.0**|
|SimCLR|Syn|63.7|72.7|70.8|
|SimCLR|Syn+Real|**64.1**|**74.9**|71.9|
|CLIP|Real|70.3|81.2|86.7|
|CLIP|Syn|67.8|79.1|83.7|
|CLIP|Syn+Real|**73.3**|**83.0**|**87.3**|
|StableRep|Syn|67.9|78.0|81.4|
|StableRep|Syn+Real|**69.1**|**79.3**|**84.2**|

To sum up, our method has significantly improved the fairness of the learned representations and the understanding of compositionality. Besides, our method shows strong generalization ability in fine-tuning setup (including dense prediction), which is surprising since it may potentially change the previously widely-accepted story that contrastive methods do not do well in fine-tuning.

Lastly, these results are achieved with **only synthetic** images.

---

### Decision · Program_Chairs · 2023-09-21

**Decision:**

Accept (poster)

**Comment:**

This submission was reviewed by five knowledgeable referees, who raised concerns about novelty, missing experimental details, a potential data poisoning, the observed performance gap on CIFAR-10/100, and the limits of generative models. The reviewers also raised important questions such as the effect of the choice of generative model on the performance, and the amount of data used to pretrain stable diffusion. The rebuttal and subsequent discussion between authors and reviewers adequately addressed the reviewers concerns. After discussion, reviewers agree that the paper presents a valuable contribution and they all recommend to accept. The AC agrees with the reviewers' assessment and therefore recommends to accept.